

# The composition, geography, biology and assembly of the coastal flora of the Cape Floristic Region

B. Adriaan Grobler and Richard M. Cowling

African Centre for Coastal Palaeoscience, Nelson Mandela University, Gqeberha, Eastern Cape Province, South Africa

Corresponding author
B. Adriaan Grobler,
adriaan.grobler85@gmail.com

## ABSTRACT

The Cape Floristic Region (CFR) is globally recognized as a hotspot of plant diversity and endemism. Much of this diversity stems from radiations associated with infertile acid sands derived from sandstones of the geologically ancient Cape Fold Belt. These ancient montane floras acted as the source for most subsequent radiations on the Cape lowlands during the Oligocene (on silcretes) and Mio–Pliocene (on shales). The geomorphic evolution of the CFR during the Plio–Pleistocene led to the first large-scale occurrence of calcareous substrata (coastal dunes and calcarenites) along the Cape coast, providing novel habitats for plant colonization and ensuing evolution of the Cape coastal flora—the most recent diversification event in the Cape. Few studies have investigated the CFR's dune and calcarenite floras, and fewer still have done so in an evolutionary context. Here, we present a unified flora of these coastal calcareous habitats of the CFR and analyze the taxonomic, biological and geographical traits of its component species to gain insights into its assembly. The Cape coastal flora, comprising 1,365 species, is taxonomically dominated by the Asteraceae, Fabaceae and Iridaceae, with *Erica*, *Aspalathus* and *Agathosma* being the most speciose genera. In terms of growth-form mix, there is a roughly equal split between herbaceous and woody species, the former dominated by geophytes and forbs, the latter by dwarf and low shrubs. Species associated with the Fynbos biome constitute the bulk of the flora, while the Subtropical Thicket and Wetland biomes also house a substantial number of species. The Cape coastal flora is a distinctly southern African assemblage, with 61% of species belonging to southern African lineages (including 35% of species with Cape affinity) and 59% being endemic to the CFR. Unique among floras from the Cape and coastal Mediterranean-climate regions is the relatively high proportion of species associated with tropical lineages, several of which are restricted to calcareous substrata of the CFR. The endemic, calcicolous component of the flora, constituting 40% of species, represents 6% of the Cape's regional plant diversity—high tallies compared to other biodiversity hotspots. Most coastal-flora endemics emerged during the Plio–Pleistocene as a product of ecological speciation upon the colonization of calcareous substrata, with the calcifugous fynbos floras of montane acid substrata being the most significant source of this diversification, especially on the typically shallow soils of calcarenite landscapes. On the other hand, renosterveld floras, associated with edaphically benign soils that are widespread on the CFR lowlands, have not been a major source of lineages to the coastal flora. Our findings suggest that, over and above the strong

pH gradient that exists on calcareous substrata, soil depth and texture may act as important edaphic filters to incorporating lineages from floras on juxtaposed substrata in the CFR.

# INTRODUCTION

The coastal flora of the Cape Floristic Region (CFR), comprising species associated with calcareous substrata (coastal dunes and calcarenites), is the youngest manifestation of the immense radiation of the Cape flora. The endemic component of this coastal flora started diversifying only in the Plio–Pleistocene, when large tracts of calcareous substrata were exposed during numerous sea-level regressions on the Palaeo-Agulhas Plain (*Marean, Cowling & Franklin, 2020*) and its equivalent on the west coast (*Dingle & Rogers, 1972*; *Cowling, Proches & Partridge, 2009*; *Hoffmann, Verboom & Cotterill, 2015*; *Cawthra et al., 2020*). These calcareous lithologies comprised unconsolidated (mainly dune sands) and consolidated (mainly calcarenites) substrata, very different chemically from most CFR soils, which are acidic. Most biologists refer to these consolidated calcareous substrata—calcarenites—as "limestones", hence the term "limestone fynbos" (*Cowling & Heijnis, 2001*; *Rebelo et al., 2006*).

The oldest radiations in the CFR are associated with the infertile, acidic, sandy soils derived from the Cape Supergroup rocks (mainly quarzitic sandstone) that form the mountains of the Cape Fold Belt. These ancient floras, intimately associated with infertile montane habitats, acted as the source flora for most subsequent radiations on the lowlands during the Cenozoic, namely on silcretes (Oligocene) and shales (Mio–Pliocene) (*Verboom, Linder & Stock, 2004*; *Cowling, Proches & Partridge, 2009*; *Verboom et al., 2014*; *Hoffmann, Verboom & Cotterill, 2015*). While coastal dunes were probably present throughout the Cenozoic, the geomorphic evolution of the CFR during the Plio–Pleistocene resulted in what was likely the first large-scale occurrence of calcareous substrata along the Cape coast, providing novel habitats for plant colonization and ensuing diversification (*Linder, 2003*; *Cowling, Proches & Partridge, 2009*).

As is the case with other nutritionally unusual or regionally rare substrata (*Kruckeberg & Rabinowitz, 1985*; *Kruckeberg, 1986*, *2002*; *Rajakaruna, 2018*), colonizing calcareous substrata would have posed physiological challenges for the Cape flora (*Thwaites & Cowling, 1988*; *Deacon, Jury & Ellis, 1992*; *Verboom, Stock & Cramer, 2017*), producing an assemblage characterised by high edaphic endemism (*Cowling, 1983*; *Cowling, Holmes & Rebelo, 1992*; *Cowling & Holmes, 1992b*; *Willis, Cowling & Lombard, 1996*; *Cowling et al., 2019*). On the other hand, CFR coastal habitats were linked directly to the coastal habitats of the summer-rainfall subtropical east coast (*Cowling, 1983*) and the winter-rainfall desert on the west coast of southern Africa (*Jürgens, 1997*), enabling colonization by extra-Cape lineages during climatically suitable times in the

Plio–Pleistocene. Furthermore, contemporary dune habitats incorporate floristic elements from abutting vegetation formations (forest, fynbos, grassland, renosterveld, succulent karoo and subtropical thicket) throughout the CFR (*Bergh et al., 2014*). As a result, we expect the coastal flora to have a strong representation of Cape lineages (fynbos, renosterveld elements) but augmented by species with desert (mainly succulent-karoo elements) and tropical (mainly subtropical-thicket and forest elements) affinities.

While there has been some research on the characteristics of the CFR's dune and calcarenite floras, these have been patchy. Aspects of the calcarenite flora, centred on the Agulhas (*Thwaites & Cowling, 1988*) and Riversdale (*Rebelo et al., 1991*) coastal plains, have been studied in part by *Cowling & Holmes (1992a, 1992b)*, *Cowling, Holmes & Rebelo (1992)* and *Willis, Cowling & Lombard (1996)*, though no complete assessment of the flora has been undertaken. The distinctiveness of the calcarenite flora is clear from the work of *Cowling (1990)*, who demonstrated nearly complete replacement of species assemblages between climatically and topographically similar sites on calcarenite and non-calcareous substrata of the Agulhas Plain. Further support for this is the recognition of the core area occupied by this flora as a centre of species endemism—termed the Bredasdorp–Riversdale (*Cowling, Holmes & Rebelo, 1992*) or Agulhas Plain centre (*Manning & Goldblatt, 2012a*)— for various Cape lineages (*e.g.*, *Dahlgren, 1963*; *Nordenstam, 1969*; *Rebelo & Siegfried, 1990*). Among substrata found on the Agulhas coastal plain, calcarenites harbour exceptionally high proportions of range-restricted edaphic endemics (*Cowling & Holmes, 1992b*), much more so than matched sites in climatically similar southwestern Australia (*Cowling et al., 1994*). *Willis, Cowling & Lombard (1996)* identified 110 calcarenite-endemic species in the calcarenite flora of the Agulhas and Riversdale coastal plains; these are especially well-represented in the Asteraceae, Ericaceae, Fabaceae and Rutaceae, but also include members of the Iridaceae and Proteaceae, and among genera, *Agathosma*, *Aspalathus*, *Erica* and *Muraltia* comprised most endemics. While the calcarenite flora is a unique assemblage of species, it shares the prominence of these higher taxa with other CFR floras.

As with the calcarenite flora, no integrated study of the Cape dune flora exists. Most botanical research has focused on strand and hummock-dune plant communities (*Boucher & Le Roux, 1993*; *Taylor & Boucher, 1993*; *Lubke et al., 1997*), with few studies exploring the floras of vegetated back dunes (*Cowling, 1983*, *1984*). *Cowling et al. (2019)* provided an analysis of a coastal dune flora from the year-round rainfall southeastern CFR, with comparisons to two winter-rainfall dune floras from the southwestern CFR. These dune floras have similar trait profiles and are dominated by members of the Asteraceae, Fabaceae and Poaceae, while families that are speciose in inland fynbos floras, such as Ericaceae and Restionaceae, are poorly represented, or entirely lacking, as is the case for Proteaceae. Typical Cape genera that are endemic or near-endemic to the Greater CFR (cf. *Born, Linder & Desmet, 2007*; cf. *Colville et al., 2014*) contribute a high proportion of species to dune floras in the region. These floras further exhibit a high frequency of species endemic to the CFR (ca. 40%) and high levels of edaphic endemism (ca. 30–40% dune endemics), although both geographic and edaphic endemism are more pronounced in the winter-rainfall zone, especially among typical Cape lineages (*sensu Linder, 2003*).

A peculiar feature of CFR dune floras is the prominence of species associated with tropical lineages, a pattern that persists even along the strongly winter-rainfall west and southwest coasts (*e.g.*, *Boucher & Jarman, 1977*), though the number of species associated with tropical lineages declines from east to west along a gradient of increasing winter rainfall (*Cowling, 1983*; *Tinley, 1985*; *Cowling et al., 1997b*; *Vlok, Euston-Brown & Cowling, 2003*). This richness of tropical species, many of which are endemic to coastal dunes of the CFR, sets Cape dune floras apart from those in other Mediterranean–climate ecosystems (MCEs) (*Cowling et al., 2019*) and from other floras in the CFR (*Cowling, 1983*, *1984*).

In summary, both calcarenites and coastal dunes in the Cape support floras that are typical of the region, yet there are apparent differences: calcarenite floras exhibit a relatively stronger Cape signature, while dunes have a strong representation of tropical elements. Despite these differences, floras associated with these substrata likely share ecological adaptations to their coastal and edaphically unique environment and experienced a common evolutionary history vastly different from that experienced by inland floras of the CFR. We therefore expect certain idiosyncrasies to emerge that may shed further light on the most recent diversification event in the Cape flora. Here, we present a unified flora of these coastal calcareous habitats of the CFR and analyse the flora to assess its size, taxonomic composition, growth-form mix, biological traits, biogeographic affinities and endemism. We further sketch a brief scenario of the flora's assembly, with a focus on the Pleistocene—a period whose dynamic sea levels and vacillating climate had a profound impact on the geography and diversification of floras in the Cape (*Cowling et al., 2017*; *Forest, Colville & Cowling, 2018*; *Colville et al., 2020*), especially along the coast (*Grobler et al., 2020*).

We find that the Cape coastal flora is a diverse assemblage of 1,365 species, with high levels of edaphic and geographic endemism at various regional scales. Similar to other floras in the CFR, the coastal flora is taxonomically dominated by the Asteraceae, Fabaceae and Iridaceae and with Cape lineages (*sensu Linder, 2003*) like *Erica*, *Aspalathus*, and *Agathosma* being the most speciose genera. However, extra-Cape lineages from desert and tropical floras have contributed several members to the coastal flora in the CFR, with many of these species being endemic to the region. The calcicolous component, representing 40% of the coastal flora, appears to largely be the product of ecological speciation following the colonization of novel calcareous substrata during the Plio–Pleistocene. The ancient, calcifuge fynbos flora, associated with acidic sands of the Cape Fold Belt, emerges as the most significant source of lineages to the coastal flora, while few lineages and species are shared with renosterveld floras found on neutral loams.

## Study area

Our study area comprised coastal dune and calcarenite landscapes of the CFR (Fig. 1). We used the delimitation of the CFR proposed by *Colville et al. (2014)*, which includes the Cape Fold Belt mountains east of Algoa Bay and the coastal belt east of here to the Kei River. This region covers an area of ca. 106,000 km$^2$. Coastal calcareous substrata occupy ca. 4,500 km$^2$ (4%) of the CFR, with areal coverage shared nearly equally between dunes (2,200 km$^2$) and calcarenites (2,300 km$^2$) (adapted from South African Council for
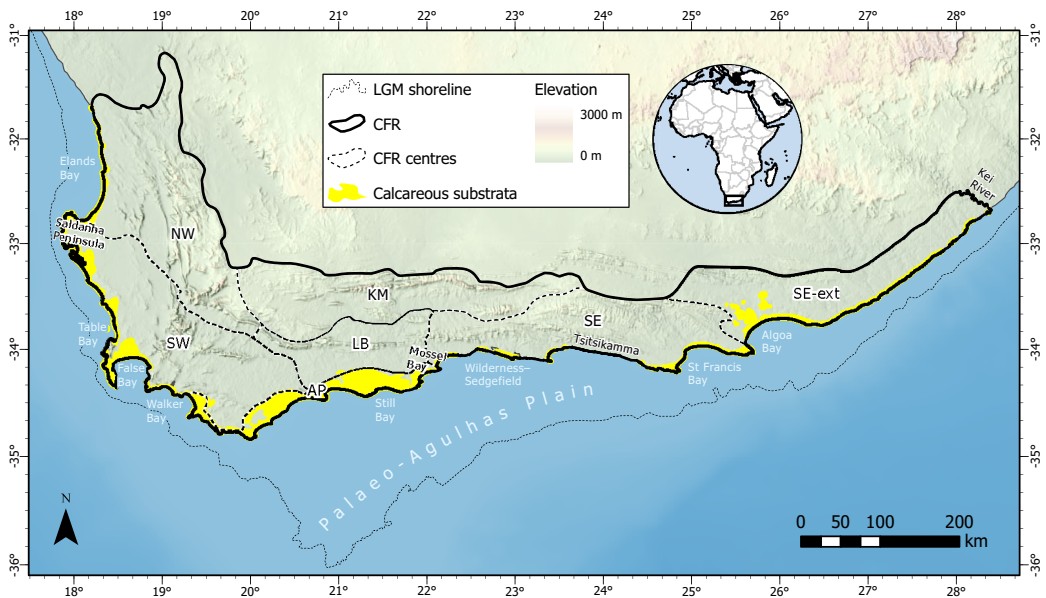

**Figure 1 Distribution of calcareous substrata (coastal dunes and calcarenites) along the southwestern, southern and southeastern coasts of South Africa with which the coastal flora of the Cape Floristic Region (CFR) is associated.** Dunes occur along the entire CFR coast, while significant calcarenite outcrops are restricted to areas along the west coast (Saldanha Peninsula), south coast (Agulhas Plain) and southeast coast (Algoa Bay). Note that the extent of calcareous substrata is slightly exaggerated for visibility (adapted from South African Council for Geoscience 1:250,000 geological database). CFR centres are after *Manning & Goldblatt (2012a)*: NW, Northwest; SW, Southwest; KM, Karoo Mountain; LB, Langeberg; SE, Southeastern; SE-ext., extension of the Southeastern centre (cf. *Colville et al., 2014*; cf. *Bradshaw, Colville & Linder, 2015*). The approximate shoreline during the Last Glacial Maximum (LGM) is taken as the 130 m isobath and indicates the extent of the Palaeo-Agulhas Plain (*Marean, Cowling & Franklin, 2020*) and analogous offshore land areas exposed at lower sea levels around 26.5 ka.

Geoscience 1:250,000 geological database). These substrata mantle most of the coastal margin, except in areas of steep coastal terrain (*e.g.*, the Tsitsikamma coast), which are not conducive to the accumulation of coastal aeolianites (*Roberts et al., 2006*). The CFR coastal lowlands were subjected to repeated marine transgressions and regressions throughout the Neogene, and especially during the interglacials and glacials of the Pleistocene (*Partridge & Maud, 2000*). During Pleistocene lowstand sequences, vast areas of calcareous substrata were exposed on the Palaeo-Agulhas Plain (*Marean, Cowling & Franklin, 2020*) along the southern margin of the contemporary CFR (*Cawthra et al., 2020*) (Fig. 1). The long-term exposure of these markedly more expansive calcareous habitats and the recurrent disturbance associated with sea-level fluctuations during the Pleistocene have likely had a profound impact on the evolution of the contemporary coastal flora in the Cape (*Grobler et al., 2020*).

Coastal dunes occur along the entire CFR coast, but are best developed along the south coast where they are associated with embayments, especially Walker Bay, Still Bay, the Wilderness–Sedgefield embayment, St Francis Bay and Algoa Bay (*Tinley, 1985*; *Roberts et al., 2006*). South-coast dunes are typically broad, high, vegetated, unidirectional parabolic dunes that form shore-parallel cordons (also known as "barrier dunes").

**Table 1 Characteristics of broad soil groups occurring on coastal lowlands of the Cape Floristic Region, corresponding to edaphic categories to which species in the coastal flora were assigned.**

| Associated geology | Texture | Fertility | pH | Depth* | Soil description | Vegetation |
|---|---|---|---|---|---|---|
| Unconsolidated coastal dunes | Sand | Moderate | Alkaline | Deep | Deep alkaline moderately fertile sand | Dune fynbos–thicket mosaic |
| Calcarenites and calcretes | Sand | Moderate | Alkaline | Shallow | Shallow alkaline moderately fertile sand | Limestone fynbos |
| Windblown cover sands | Sand | Low | Acidic | Deep | Deep acidic low-fertility sand | Sand fynbos |
| Quartzitic sandstones | Sand | Low | Acidic | Shallow | Shallow acidic low-fertility sand | Sandstone fynbos |
| Shales, mudstones, conglomerates | Loam | High | Neutral | Deep | Deep neutral highly fertile loam | Renosterveld |

Notes:
* Deep, > 0.3 m; Shallow, < 0.3 m.
Note that these categories were not mutually exclusive (*i.e.*, certain species occur on multiple soil types).

The monoclinal southwest and southeast coasts, on the other hand, generally host smaller dunes: plumes of low, vegetated hairpin dunes, often extending some distance inland, predominate in the southwest (Elands Bay to Table Bay and False Bay), while a narrow cordon of densely vegetated, bidirectional parabolic dunes is typical of the far southeast (east of Algoa Bay) (*Tinley, 1985*; *Roberts, Cawthra & Musekiwa, 2014*). Most extant coastal dunes are geologically young, with their deposition precipitated by rising sea levels since the terminal Pleistocene and start of the Holocene (*Roberts, Cawthra & Musekiwa, 2014*). Our interest lies in these young dunes and not older Neogene dunes that have been subjected to leaching and oxidization through long-term weathering (*Tinley, 1985*) and whose floras are therefore similar to non-calcareous, inland habitats of the CFR (*Cowling & Holmes, 1992a*).

While calcarenites are present along most of the Cape coastal margin (*Brooke, 2001*), the most significant exposures occur on the Agulhas and Riversdale coastal plains of the south coast, with smaller outcrops on the southwest coast around the Saldanha Peninsula, and on the southeast coast inland of Algoa Bay (*Roberts et al., 2006*; *Roberts, Cawthra & Musekiwa, 2014*) (Fig. 1). These formations and their associated colluvial deposits form distinctive relief features, especially along the south coast, where they comprise elongate, shore-parallel, transverse ridges extending up to 15 km inland (*Roberts et al., 2006*; *Roberts, Cawthra & Musekiwa, 2014*). They are interpreted as highstand deposits dating primarily from the Pliocene.

Soils associated with coastal dunes and calcarenites are similar, comprising well-drained, coarse- to medium-grained, alkaline (pH 7–8) sands with moderate levels of available phosphorous (5–20 ppm), though dune soils are mostly deep (>1 m), while soils overlying calcarenites are much shallower (<0.3 m) (*Cowling, 1984*; *Tinley, 1985*; *Thwaites & Cowling, 1988*) (Table 1). Colluvial sands that fringe calcarenite ridges are deeper (>0.3 m), moderately leached and neutral (*Thwaites & Cowling, 1988*).

The climate varies from winter-rainfall in the southwest (Elands Bay to Cape Agulhas) to a non-seasonal rainfall regime where the proportion of summer precipitation increases eastward, but with rainfall peaks in autumn and spring (*Deacon, Jury & Ellis, 1992*; *Schulze, 2008*; *Bradshaw & Cowling, 2014*). Mean annual precipitation varies from 300–800 mm, with lowest rainfall in the west around the Saldanha Peninsula (300 mm)

and along the south coast between Cape Agulhas and Mossel Bay (300–500 mm), and highest rainfall in the east toward the mouth of the Kei River (800 mm) (*Schulze, 2008*). On the semi-arid west coast, fog is an important source of moisture, especially during autumn and summer (*Rebelo et al., 2006*; *Bradshaw & Cowling, 2014*). Temperatures are generally mild with mean temperatures ranging from 18–22 °C in midsummer and 14–16 °C in midwinter, although mean daily maxima are ca. 26 °C during summer and mean daily minima ca. 8 °C during winter (*Rebelo et al., 2006*; *Schulze, 2008*). Due to the strong marine influence, frost is a rare phenomenon. Strong winds and gales are common during summer (easterly winds) and winter (westerly winds), although the central region (Mossel Bay to Tsitsikamma) generally has a calmer wind regime (*Schulze, 2008*).

Coastal dune vegetation predominantly comprises a mosaic of two biomes, namely Fynbos and Subtropical Thicket, termed 'dune fynbos–thicket mosaic' (*Cowling, 1984*; *Tinley, 1985*; *Cowling et al., 1988*; *Rebelo et al., 1991*; *Vlok, Euston-Brown & Cowling, 2003*; *Zietsman & Bredenkamp, 2006*). The former is a low, fire-prone shrubland dominated by evergreen, small-leaved shrubs occurring mostly in fire-exposed and edaphically dry sites (Fig. 2A), while the latter is a medium to high, dense, closed-canopy shrubland dominated by evergreen, large-leaved shrubs occurring in mesic, fire-protected sites (Fig. 2B). Where ample moisture is available and fire is excluded for long periods, dune thicket can attain forest stature. Grasslands are largely a feature of the southeastern CFR coast (St Francis Bay to Kei River) and become increasingly dominant on dunes east of Algoa Bay (*Vlok, Euston-Brown & Cowling, 2003*). Coastal dune landscapes further support various wetland types as well as distinctly coastal habitats, including mobile bypass dunes, semi-mobile hummock dunes above sandy shores, and semi-succulent herblands and shrublands above rocky shores.

On the south coast (False Bay to Mossel Bay), vegetation associated with calcarenites comprises a distinct Fynbos-biome formation known as 'limestone fynbos' (*Rebelo et al., 2006*). It generally comprises fire-prone, tall, evergreen shrublands dominated by overstorey proteoid shrubs (*Leucadendron*, *Leucospermum*, *Protea*) (*Cowling et al., 1988*; *Rebelo et al., 1991*) (Figs. 2C, 2D). As is the case with coastal dunes, calcarenite landscapes may also support pockets of subtropical thicket or forest in moist, fire-sheltered sites, although these are far more restricted in extent than is the case for dunes. On the semi-arid west coast around the Saldanha Peninsula, calcarenites support 'strandveld'—a low, succulent-rich, subtropical-thicket shrubland (*Boucher & Jarman, 1977*; *Rebelo et al., 2006*) (Fig. 2E). Calcarenites occurring inland of Algoa Bay on the southeast coast support a mosaic of subtropical thicket, typically occurring as small rounded clumps in dolines (*Carvalho & Campbell, 2021*), and a grassy, succulent-rich dwarf-shrubland (*Taylor & Morris, 1981*; *Vlok, Euston-Brown & Cowling, 2003*) (Fig. 2F).

As alluded to in the preceding paragraphs, and as is the case in the CFR more generally (*Kraaij & Van Wilgen, 2014*), fire plays an important role in the functioning of ecosystems occurring in dune and calcarenite landscapes of the Cape's coastal forelands. All habitats in these landscapes, other than the forest patches, wetlands, bypass dunes, hummock dunes and near-shore herb- and shrublands, are subject to wildfires at moderate

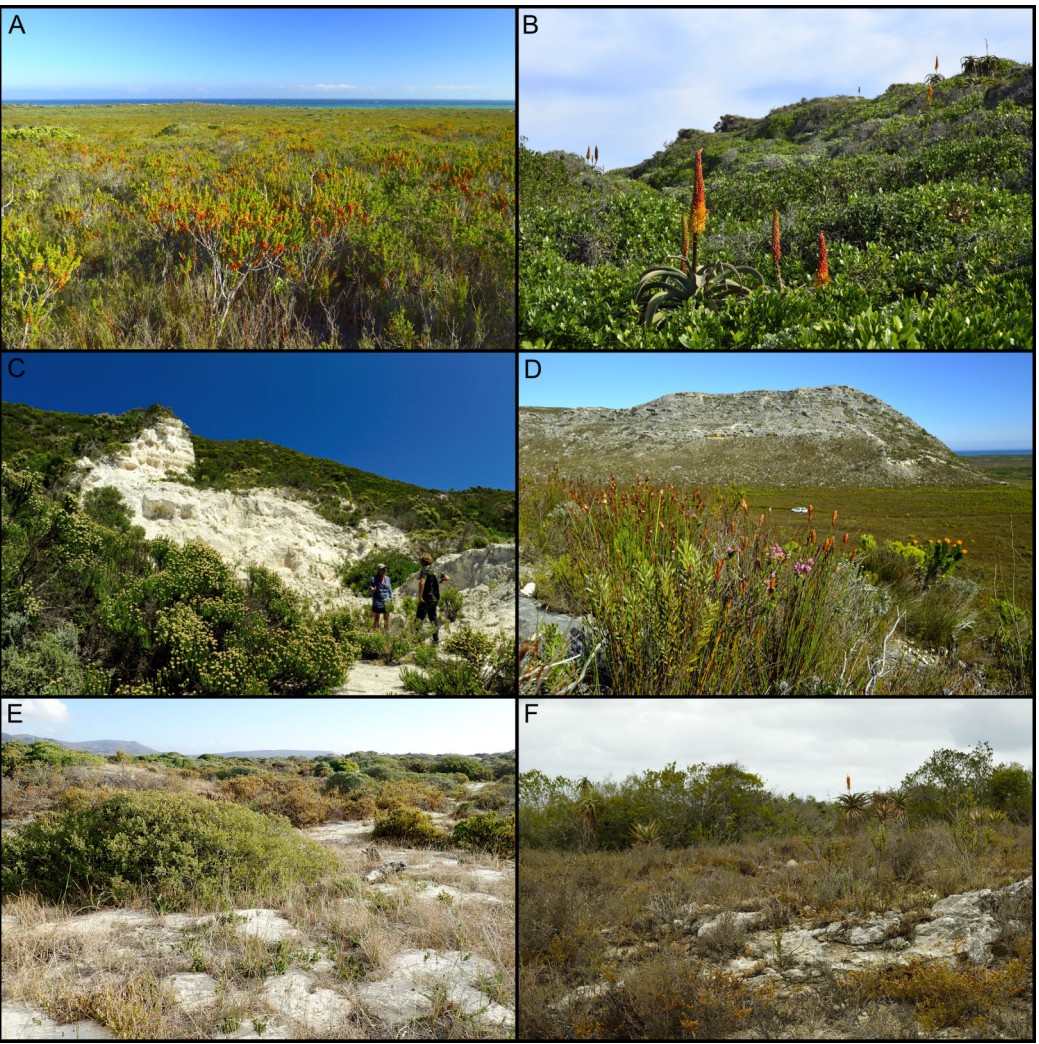

**Figure 2 Dominant vegetation formations on coastal calcareous substrata of the Cape Floristic Region.** (A) Dune fynbos–thicket mosaic near Walker Bay, with the fynbos component dominant here. (B) Dune fynbos–thicket mosaic near Algoa Bay, with the thicket component dominant here. (C, D) Limestone fynbos on calcarenites near Walker Bay. (E) Strandveld, occurring on coastal dunes and calcarenites along the semi-arid west coast, near False Bay. (F) Grassy-shrubland–thicket mosaic occurring on calcarenites inland of Algoa Bay. Photographs by B. Adriaan Grobler (A–D, F) and Richard M. Cowling (E).     

intervals (10–30 years) (*Cowling et al., 1997a*, *2019*). The fire ecology of calcarenite grassy-shrubland–thicket mosaics inland of Algoa Bay is poorly understood, but the incidence of fire-dependent Cape lineages (*e.g.*, Diosmeae, *Muraltia*, Restionaceae) (*sensu Linder, 2003*) in the matrix vegetation (*Taylor & Morris, 1981*) and the prominence of *Pterocelastrus tricuspidatus*—a strong post-fire resprouter (*Strydom et al., 2020*)—in thicket clumps (*Taylor & Morris, 1981*; *Carvalho & Campbell, 2021*) suggest that fire is a periodic disturbance in this system. *Kraaij et al. (2020)* have proposed that historical fire regimes on coastal lowlands in the southern Cape supported fire-prone and -dependent vegetation throughout the Pleistocene, including dune fynbos–thicket mosaics

and limestone fynbos analogous to those of the contemporary CFR (*Cowling et al., 2020*). Thus, as in the CFR more generally (*Bytebier et al., 2011*; *He, Lamont & Manning, 2016*; *Rundel et al., 2018*), fire has been an important factor shaping the evolution of the Cape's coastal flora.

## METHODS

### Flora compilation

Our flora (including only native flowering plant species) of coastal calcareous substrata was compiled from a variety of sources including scientific literature, online databases and personal photographic records. The primary source was the most recent plant conspectus for the CFR (*Manning & Goldblatt, 2012a*). However, the authors of this work used a narrower circumscription of the CFR than we do (cf. *Colville et al., 2014*), essentially excluding the coastal mountains and lowlands between Algoa Bay and the Kei River. We therefore also referred to the most recent plant conspectus for the Eastern Cape Province (*Bredenkamp, 2019*), which encompasses this eastern coastal extension of the CFR. Both floras include habitat descriptions for plant species, which we queried for relevant terms, namely "alkaline", "beach", "calcareous", "coastal", "dune", "limestone" and "strand". The term "coastal" often included species not present on calcareous substrata, which we excluded from our flora. These are species associated with leached, siliceous cover sands that blanket patches of coastal foreland—typically inland of the calcareous substrata—on both the west and the south coasts of the CFR. Thus, the floras associated with Sand Fynbos vegetation types (cf. *Rebelo et al., 2006*) were not included. We also excluded species that occur only east of Algoa Bay (the Sundays River), typically of tropical affinity and associated with coastal forests (*e.g.*, *Eugenia capensis*, *Mimusops obovata*, *Strelitzia nicolai*).

The list of species derived from these two conspectuses were supplemented with floras for coastal dune habitats produced by *Olivier (1983)*, *Van Wijk et al. (2017)* and *Cowling et al. (2019)*, and by incidental species lists produced during vegetation surveys in coastal dune and calcarenite habitats (*Zietsman & Bredenkamp, 2006*, *2007*; *Mergili & Privett, 2008*). While no structured field work was undertaken for this study, we also included georeferenced photographic records of species collected by us from 2009–2021. Species were identified by us from photographs and these photographic records (1,631 records of 506 species) were collated in an online biodiversity records database, iNaturalist (https://www.inaturalist.org/observations?q=CFR_coastal_flora&user_id=adriaan_grobler), where they were (and are) available for scrutiny by other botanists. Select species records from other iNaturalist contributors, which were manually inspected and verified by us, were also included.

We further verified the edaphic occurrence of species for key taxonomic groups from various descriptions, revisions and monographs (*Levyns, 1954*; *Nordenstam, 1968*; *Grau, 1973*; *Verdoorn, 1980*; *Puff, 1986*; *Linder, 1990*; *Schubert & Wyk, 1997*; *Linder & Mann, 1998*; *Whitehouse, 2002*; *Van Jaarsveld & Koutnik, 2004*; *Goldblatt & Manning, 2007*, *2010*, *2011*; *Manning, Goldblatt & Forest, 2009*; *Manning & Goldblatt, 2010*, *2012b*; *Köcke et al., 2010*; *Nkonki, 2013*; *Wolfe, 2013*; *Fish et al., 2015*; *Bello et al., 2017*; *Stirton & Muasya, 2017*;

*Bergh & Manning, 2019*; *Manning, 2019*; *Alonso et al., 2021*), adding those species that were not included during the previous steps. As a final verification, two speciose families in our flora were checked by taxonomic experts to assess the validity of species we included (Ericaceae, checked by Ross C. Turner; Rutaceae, checked by Terry H. Trinder-Smith). These experts also verified traits of species, as described below, for their focal taxa. We included putatively undescribed species in our flora; where these were derived from our own observations, we communicated with experts to confirm their taxonomic placement (Charles H. Stirton, *Otholobium* sp. nov. 'algoensis'). The primary source for each species included in our flora is indicated in Data S1. Nomenclature in our flora follows the Plants of Southern Africa database provided by the South African National Biodiversity Institute (http://posa.sanbi.org).

## Biological traits

Species were categorized into five herbaceous and five woody growth forms. Classifications were based on information from the literature (listed under "Flora compilation") and from our observations in the field. In the case of species with high phenotypic plasticity, we assigned to it the growth form that is most prevalent throughout our study area. As an example, *Sideroxylon inerme*, which takes many forms—from a dwarf multi-stemmed shrub to a tall single-stemmed tree—most commonly grows as a tall, multi-stemmed shrub in our study area and was categorized as such. Herbaceous growth forms comprised evergreen hemicryptophytes, deciduous hemicryptophytes, geophytes, annuals, forbs and vines. Woody growth forms were trees (mostly single-stemmed, >5 m height), tall shrubs (2–5 m height), low shrubs (0.5–2 m height), dwarf shrubs (<0.5 m height) and lianas. Herbaceous and woody species exhibiting succulence (leaf or stem), and putatively employing CAM photosynthesis (*Mooney, Troughton & Berry, 1977*), were additionally classified as succulents.

We further classified woody species according to their postfire-regeneration mode, using a simplified version of *Pausas et al. (2016)*'s schema to match the resolution of information available to us. Species were categorized as obligate resprouters (plants exclusively resprout after fire, recruit from seeds in favourable microsites during fire-free intervals), facultative seeders (plants capable of resprouting and establishing seedlings after fire) and non-sprouters (plants killed by fire and reliant on seeds for regeneration, including obligate seeders and postfire colonizers), based in part on the literature (*Cowling & Pierce, 1988*; *Strydom et al., 2020*; works listed under "Flora compilation"), but largely following our own observations after several wildfires at various sites along the Cape south coast between 2016 and 2021. Regeneration modes of *Aspalathus* species were gleaned from supplemental data provided by *Cowling et al. (2018)*.

## Geographical traits

Using habitat descriptions from the literature (listed under "Flora compilation") and based on our observations throughout the study area, species were allocated to one of the following biomes: Fynbos, Subtropical Thicket (hereafter 'Thicket'), Forest, Grassland,

Coastal, Wetland or Disturbed. While most species have a strong affinity for a specific biome, others may occur in multiple biomes (*e.g.*, *Pterocelastrus tricuspidatus*, which is found in Thicket and Forest); in such cases, species were assigned to the biome in which they are most prevalent throughout the study area (Thicket, in the case of *P. tricuspidatus*).

Species were further categorized according to the distribution of their respective genera in relation to phytogeographic regions, based on information from *Goldblatt (1978)*, *Manning & Goldblatt (2012a)*, and *Bredenkamp (2019)*. A species was assigned to one of seven regions to which its genus is endemic or near-endemic. The seven regional categories used were: Greater Cape Floristic Region (GCFR—comprising the CFR and adjacent winter-rainfall semi-desert region); Southern Africa (roughly corresponding to the countries of South Africa, Lesotho, Eswatini, Botswana and Namibia, and thus including the GCFR as a subset); Afrotemperate (disjunct montane regions embedded in Southern Africa and the Afrotropical region, extending from the CFR northward through the Drakensberg range to the East African Rift System and the highlands of Cameroon, Guinea and Ethiopia); Afrotropical (roughly corresponding to sub-Saharan Africa, including tropical and subtropical enclaves in Southern Africa, but excluding the Afrotemperate region); Pantemperate (temperate regions beyond Africa); Pantropical (tropical regions beyond Africa); and Cosmopolitan (globally widespread).

We also classed species based on their distribution within the GCFR, identifying those endemic to: the entire region; only the CFR; and those with distributions extending beyond the GCFR (*i.e.*, non-endemic). Additionally, species endemic to single centres within the CFR (Agulhas Plain Centre, Southeastern Centre, Southwestern Centre; Fig. 1) were identified, as were local-endemic species. We define the latter as those species that are restricted to a single CFR centre with a lower-than-average range size in which the extent of occurrence is typically <2,000 km$^2$ (*Cowling, 2001*). Note that select species included in this category have extremely small range sizes (<5 km$^2$) and may be regarded as point endemics.

## Edaphic traits

Species were assigned to edaphic categories corresponding to four broad, biologically relevant soil types found on the coastal lowlands of the CFR (Table 1) (cf. *Deacon, Jury & Ellis, 1992*; cf. *Cawthra et al., 2020*; cf. *Cowling et al., 2020*). Groups are based on soil texture, fertility, pH and depth, and include: (1) moderately fertile, alkaline sands associated with coastal dunes (typically deep) and calcarenites (typically shallow); (2) shallow, acidic, infertile sands associated with quartzitic sandstones; (3) deep, acidic, infertile windblown cover sands; and (4) deep to shallow, neutral, fertile loams associated with shales, mudstones and conglomerates. These categories were not mutually exclusive, with some species occurring on multiple soil types. Species occurring on alkaline sands were also scored for edaphic endemism on coastal dunes and on calcarenites. Again, these were not mutually exclusive, with some calcicoles occurring on both calcareous substrata (but being restricted to them).

## Data analysis

All data analysed in this study are available in Data S1. We performed our analyses using R version 4.1.0 statistical software (*R Core Team, 2021*), with additional use of the 'tidyverse' version 1.3.1 (*Wickham et al., 2019*), 'janitor' version 2.1.0 (*Firke, 2021*), 'treemapify' version 2.5.5 (*Wilkins & Rudis, 2021*), 'igraph' version 1.2.6 (*Csardi & Nepusz, 2006*), ggraph version 2.0.5 (*Pedersen, 2021*) and 'patchwork' version 1.0.1 (*Pedersen, 2020*) R packages for data cleaning, wrangling, analysis and visualization.

# RESULTS

## Floristic composition

The coastal flora of the CFR comprises 1,365 species, 435 genera and 102 families (the full list of species is presented in Data S1). The largest families are the Asteraceae (198 spp.), Fabaceae (103 spp.), Iridaceae (76 spp.), Rutaceae and Scrophulariaceae (both 61 spp.), Aizoaceae (59 spp.), Poaceae (53 spp.) and Cyperaceae (47 spp.) (Fig. 3; Table S1). Together, these eight families—representing less than a tenth of the coastal flora families—account for nearly half of the species (658 spp.), while the 20 most speciose families (a fifth of coastal flora families) contribute nearly 70% of species (946 spp.) in the flora. In contrast, half (52 families) of the recorded families comprise fewer than five species, with most of these (27 families) represented by a single species. Thus, relatively few families account for the bulk of species in the Cape coastal flora, while most families contribute only a few species.

Patterns of family size remain similar when comparing numbers of genera (Fig. 3; Table S1): the Asteraceae contributes the most genera (53 genera) to the coastal flora, while the Poaceae (30 genera), Fabaceae (22 genera), Aizoaceae (19 genera), Iridaceae (16 genera), Cyperaceae (14 genera) and Scrophulariaceae (13 genera) are also genus-rich. The Apiaceae, while being relatively species-rich (29 spp.), ranks higher among families for generic richness (15 genera), and the Rutaceae, one of the most speciose families, ranks lower (nine genera). Most families (90 families) are represented by fewer than 10 genera, with most of these (61 families) comprising only one or two genera. Families represented by a single genus total 47.

The most speciose genera in the Cape coastal flora are *Erica* and *Aspalathus* (both 28 spp.), *Agathosma* (26 spp.), *Senecio* (25 spp.), *Helichrysum* (24 spp.), *Indigofera* (23 spp.) and *Hermannia* (22 spp.) (Fig. 3; Table S1). These genera represent less than 2% of genera recorded in the coastal flora but comprise 13% (176 spp.) of species. The 20 largest genera (5% of coastal flora genera) account for 27% of species (366 spp.), while nearly half (216 genera) of the genera in the coastal flora are represented by a single species.

The overall ratio of species to genera (S/G-ratio) in the Cape coastal flora is 3.1, with most families (73 families) in the flora having an S/G-ratio lower than this. For nearly a third of coastal flora families (29 families), the S/G-ratio is higher than that of the whole flora (Table S1). Most notable are the Ericaceae (28.0), Polygalaceae and Asparagaceae (both 13.0), and Oxalidaceae (12.0), though importantly, these families all contribute only one (Asparagaceae, Ericaceae, Oxalidaceae) or two (Polygalaceae) genera

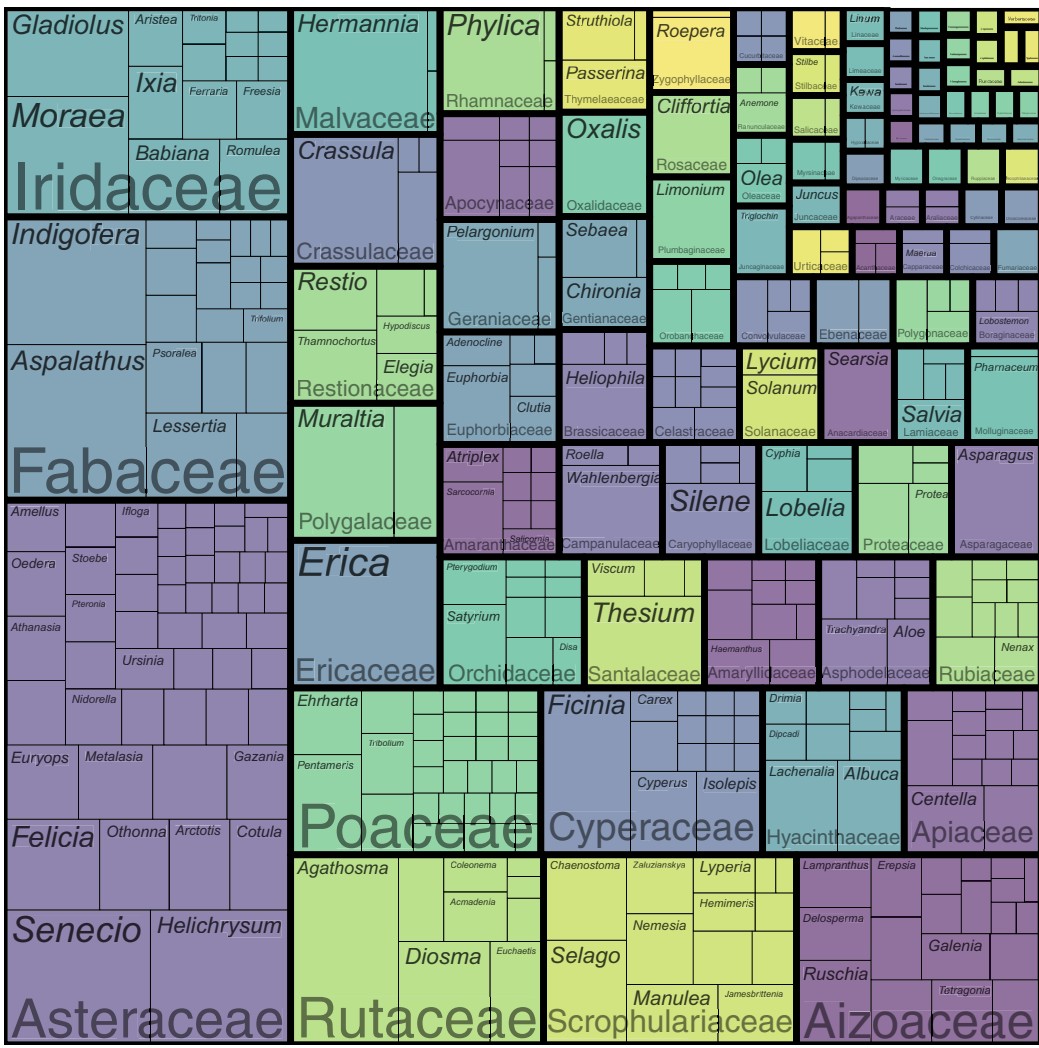

**Figure 3  Treemap of the coastal flora of the Cape Floristic Region.** Depicted are the relative number of species per genus per family. The total flora comprises 1,365 species. Note that colours only aid to differentiate different plant families.              

to the flora. Of the ten most speciose coastal flora families, only the Rutacaeae has a comparatively high S/G-ratio (6.8), with the ratio for the remaining nine families ranging from 1.8–4.8.

## Biological traits

Herbaceous growth forms comprise about half (49%, 667 spp.) of the flora (Fig. 4A). Among herbaceous species, geophytes are most numerous (30%, 199 spp.), with strong representation of the Iridaceae (*Moraea*, *Gladiolus*), while the Hyacinthaceae, Orchidaceae and Amaryllidaceae are also prominent in the geophytic flora. The bulk of geophytes are thus petaloid monocots, though some dicots (*Oxalis*, *Othonna*, *Pelargonium*) do occur. A quarter (25%, 165 spp.) of herbaceous species are forbs, with most being members of the Asteraceae. Hemicryptophytes constitute 21% of herbs; most of these (12% of herbs, 81 spp.) are evergreen (Cyperaceae, Restionaceae), while deciduous species (mainly

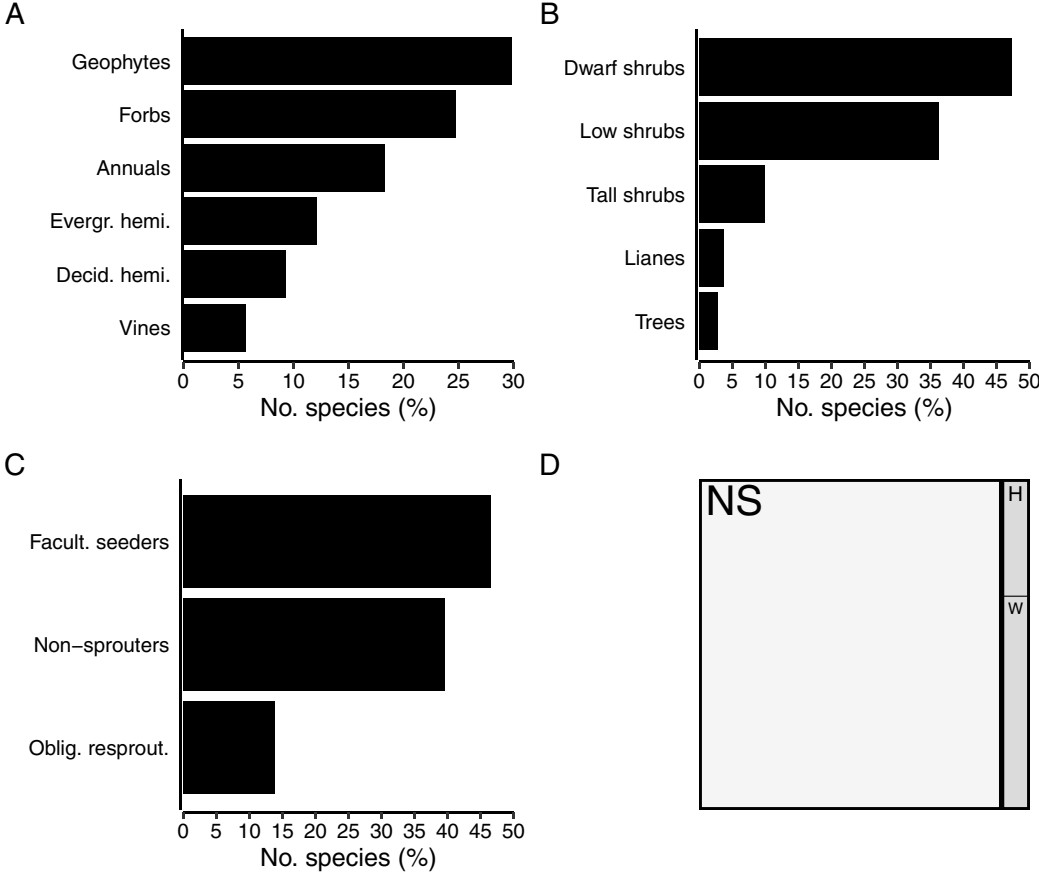

**Figure 4 Biological traits of species in the coastal flora of the Cape Floristic Region (CFR), comprising 1,365 species.** (A) Proportion of species exhibiting different herbaceous growth forms (667 spp. total). (B) Proportion of species exhibiting different woody growth forms (698 spp. total). (C) Proportion of woody species exhibiting different post-fire regeneration strategies (698 spp. total). (D) Treemap of succulence in the coastal flora, showing proportion of non-succulent (white, NS) and succulent (grey) species (116 spp.), and proportion of herbaceous (H) and woody (W) succulent species.

Poaceae) make up a slightly smaller proportion (9% of herbs, 62 spp.). About 18% of herbaceous species (122 spp.) in the coastal flora are annuals, with the annual flora dominated by the Asteraceae and Scrophulariaceae. Vines—typically associated with forest and thicket—comprise nearly 6% of herbs (38 spp.), with most of these herbaceous climbers belonging to the Apocynaceae.

Just over half of the coastal flora are woody species (51%, 698 spp.), with dwarf shrubs (47% of woody species, 330 spp.) and low shrubs (36% of woody species, 253 spp.) making up the bulk of these (Fig. 4B). Among dwarf shrubs, the Fabaceae (*Indigofera*, *Aspalathus*) is most species-rich, followed by the Aizoaceae and Asteraceae. The Asteraceae (*Helichrysum*) and Fabaceae (*Aspalathus*) also contribute several low shrubs to the coastal flora, as does the Rutaceae (*Agathosma*, *Diosma*, *Euchaetis*). About 10% of woody species are tall shrubs (69 spp.), most of which are members of the Asteraceae (*Metalasia*) but also include Proteaceae (*Leucadendron*), though several belong to typically tropical families (Celastraceae, Anacardiaceae, Ebenaceae). The tree flora—an

exclusive feature of the Forest biome—is sparse, comprising only 20 species (3% of woody species) and with no taxonomic groups being dominant. Lianas account for 4% (26 spp.) of woody species, and so climbing plants (lianas and vines) constitute nearly 5% (64 spp.) of the CFR coastal flora. The genus *Asparagus* (Asparagaceae) is especially diverse among lianas.

Nearly half (47%, 325 spp.) of woody species in the coastal flora are facultative seeders, most being associated with the Fynbos biome (Fig. 4C). Similarly, the bulk of seed-reliant non-resprouters, which make up 40% (276 spp.) of woody species, are associated with fynbos. Obligate resprouters, typically tall shrubs and trees of tropical origin associated with thicket and forest, comprise 14% (97 spp.) of woody species in the coastal flora.

Succulents, which include both herbaceous and woody growth forms, comprise 116 species (8% of the flora) (Fig. 4D). About a third of succulents are herbaceous (41 spp.), while the bulk are woody (75 spp.). Most succulent species are members of the Aizoaceae (*Ruschia, Delosperma, Lampranthus, Mesembryanthemum, Tetragonia*), but Crassulaceae (*Cotyledon, Crassula*) and Asphodelaceae (*Aloe, Bulbine*) also feature prominently. Most species are leaf-succulents, with only a few *Euphorbia* species being stem-succulents.

## Geographical traits

Species associated with the Fynbos biome constitute most of the CFR coastal flora (64%, 873 spp.) (Fig. 5A), represented mainly by the Asteraceae (*Helichrysum, Senecio*), Fabaceae (*Aspalathus, Indigofera*), Rutaceae (*Agathosma, Diosma*), Iridaceae (*Moraea, Gladiolus*) and Scrophulariaceae. The Thicket biome houses 11% (153 spp.) of the flora, with members of the Apocynaceae, Anacardiaceae (*Searsia*), Celastraceae, and Asparagaceae (*Asparagus*) well represented. A small but distinct suite of species is associated with the Forest biome, accounting for 5% (73 spp.) of the coastal flora, including mostly members of tropical genera (*Apodytes, Erythrina, Maurocenia, Psychotria*), but also Afrotemperate elements (*Afrocarpus*). The Cyperaceae, Asteraceae and Amaranthaceae dominate the Wetland biome, which hosts 8% (111 spp.) of species. The flora of the Coastal biome, hosts about 5% (69 spp.) of species, most of which belong to the Asteraceae, Aizoaceae and Poaceae. The Grassland biome—largely a feature of the southeastern CFR—comprises nearly 4% (51 spp.) of the coastal flora and is characterized by a high number of Poaceae with tropical origins ($C_4$-genera, *e.g.*, *Cymbopogon, Eragrostis, Themeda*). Ruderal species typical of disturbed areas (*e.g.*, paths, roadsides or patches of recently cleared vegetation) constitute less than 3% (35 spp.) of the flora, with the Aizoaceae (*Galenia*) and Asteraceae (*Berkheya, Senecio*) being most frequent.

Most species in the Cape coastal flora belong to 124 genera endemic or near-endemic to the Greater Cape Floristic Region (GCFR) (35%, 474 spp.) (*Aspalathus, Erica, Agathosma, Muraltia*) and 85 genera endemic to the Southern African Region (26%, 348 spp.) (*Hermannia, Ficinia, Moraea, Crassula*); more than 60% of species in the flora thus have a southern-African affinity (Fig. 5B). Cosmopolitan genera (*e.g.*, *Cyperus, Polygala, Senecio*) comprise 11% (155 spp.) of species. Temperate lineages contribute close to 15% (201 spp.) of the flora, with species shared equally between 44 pantemperate (*e.g.*, *Wahlenbergia, Silene, Limonium*) and 24 Afrotemperate genera (*e.g.*, *Helichrysum,*

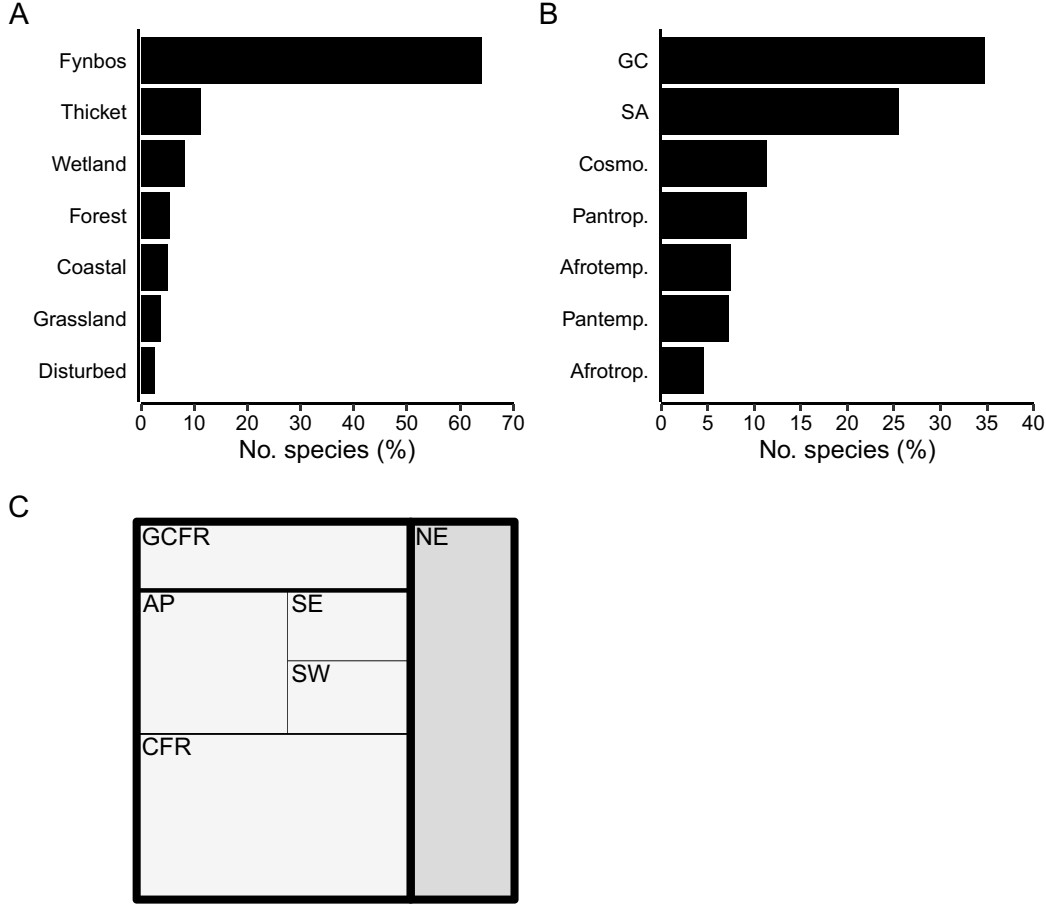

**Figure 5 Geographical traits of species in the coastal flora of the Cape Floristic Region (CFR), comprising 1,365 species.** (A) Biome association, showing the proportion of species associated with each biome occurring on coastal forelands of the CFR. (B) Generic biogeography, showing the proportion of species whose genera are associated with different phytogeographic regions: GC, Greater Cape; SA, Southern Africa; Cosmo., Cosmopolitan; Pantr., Pantropical; Afrotemp., Afrotemperate; Pantemp., Pantemperate; Afrotrop., Afrotropical. (C) Treemap of regional endemism, showing the proportion of GCFR-endemic (white) and non-endemic species (grey), as well as proportion of species endemic to hierarchical phytogeographic units in the Greater Cape Floristic Region (GCFR). CFR centres are: AP, Agulhas Plain; SE, Southeastern; SW, Southwestern. Note that proportions of species in GCFR units are cumulative (*i.e.*, total CFR endemics = CFR centres + CFR, total GCFR endemics = CFR centres + CFR + GCFR).

*Gladiolus, Albuca*). Nearly 14% of species belong to tropical lineages, most of which (9%, 125 spp.) are associated with 76 pantropical genera (*e.g.*, *Diospyros, Olea, Ipomoea*), and a smaller number (5%, 62 spp.) with 29 Afrotropical genera (*e.g.*, *Asparagus, Searsia, Maerua*).

Species endemic to the GCFR constitute about 73% (990 spp.) of the flora, most of which are CFR-endemics (810 spp., 59% of the total flora) (Fig. 5C). Many species are endemic to a single phytogeographic centre within the CFR (376 spp., 28%), with the Agulhas Plain Centre (AP) housing most of these (207 spp.), followed by the Southwestern (SW) (86 spp.) and Southeastern (SE) Centres (84 spp.). Several of these centre-endemic species (196 spp.) have notably restricted distributions (*i.e.*, local endemics), with their

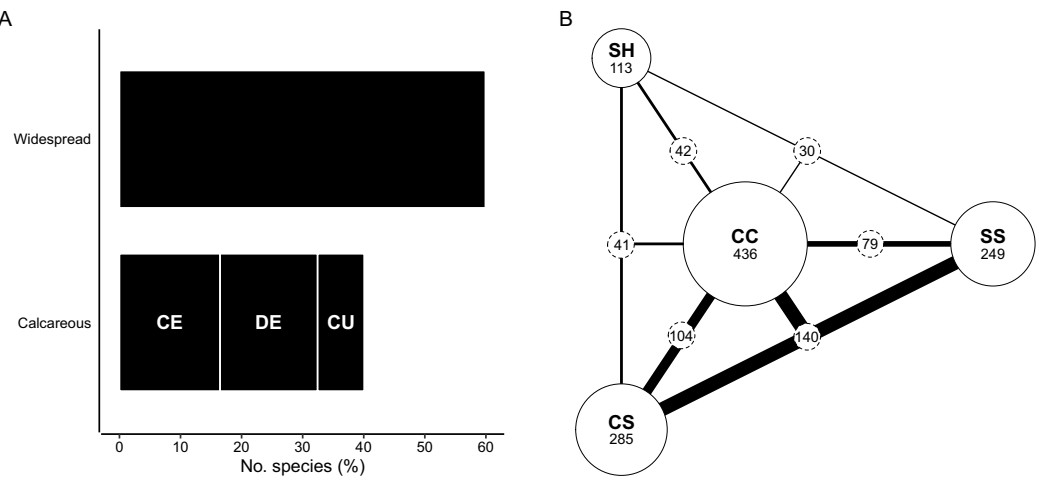

**Figure 6 Edaphic traits of species in the coastal flora of the Cape Floristic Region (CFR), comprising 1,365 species.** (A) Edaphic distribution of species in the CFR coastal flora. Widespread species occur on calcareous as well as other substrata in the CFR. Species restricted to calcareous substrata are: CE, calcarenite endemics; DE, dune endemics; CU, calcicole ubiquists occurring on both calcarenites and dunes. (B) Distribution of edaphically widespread species that exhibit some edaphic association (*i.e.*, excluding species that occur on all substrate types) on four major substrata of the CFR (436 spp. total). Substrata are: CC, alkaline moderately fertile sand (coastal dunes and calcarenites); CS, deep acidic low-fertility sand; SS, shallow acidic low-fertility sand; SH, deep neutral highly fertile loam. The size of solid circles is scaled relative to the number of species occurring on each substrate (species numbers shown under substrate codes), while the width of lines connecting substrata is scaled relative to the number of species shared between those substrata (species numbers shown in stippled circles along lines). See Table 1 for substrate characteristics.

regional richness reflecting that of centre endemics: most local endemics occur in the AP (119 spp.), followed by the SW (51 spp.) and SE (26 spp.). None of the species occurring in the coastal flora are endemic to the Northwestern Centre. The genera *Erica* (23 centre-endemic spp., 16 local-endemic spp.), *Agathosma* (18 spp., 10 spp.), *Aspalathus* (15 spp., 10 spp.), *Indigofera* (14 spp., 6 spp.), *Muraltia* (11 spp., 8 spp.) and *Phylica* (9 spp., 5 spp.) are strongly represented among range-restricted species.

## Edaphic traits

Species that are edaphically widespread—those occurring on calcareous substrata as well as other soils of the CFR—make up 60% of the flora (817 spp.) (Fig. 6A). Most of the edaphic-wides (382 spp., 28% of the flora) show no affinity for specific substrata and occur on all soil types. Of the 436 remaining edaphically widespread species (Fig. 6B), most occur elsewhere on deep, windblown acid sands (CS; 285 spp.) and/or sandstone-derived acid sands (SS; 249 spp.), while those that occur on neutral loams (SH; 113 spp.) are less common in the coastal flora. Among these species, three prominent edaphic groups emerge: (1) species typically associated with calcareous sands as well as deep and shallow acid sands (CC + CS + SS) (140 spp., 10%); (2) species occurring only on calcareous sands and deep, acid sands (CC + CS) (104 spp., 8%); and (3) species found on calcareous sands and shallow, sandstone-derived acid sands (CC + SS) (79 spp., 6%). Species shared between calcareous sands and neutral loams (CC + SH) (42 spp.), and those shared

between calcareous sands, neutral loams and windblown acid sands (CC + SH + CS) (41 spp.) comprise the same, relatively small proportion of the flora (both 3%). Edaphic-wides that occur elsewhere on neutral loams and sandstone-derived acid sands (CC + SH + SS) are least frequent in the coastal flora (30 spp., 2%).

About 40% (548 spp.) of species in the CFR coastal flora are endemic to calcareous substrata, with their occurrence shared equally among dunes and calcarenites (both 24% of total flora); of these, 104 species (8% of total flora) occur on both dunes and calcarenites (Fig. 6A). Dune-endemics total 226 species (17% of total flora), while 218 species (16% of total flora) are endemic to calcarenites (note that these tallies exclude the 104 calcicolous species occurring on both dunes and calcarenites).

## The calcicolous component

Most (52%) of the 548 calcicolous species—those restricted to calcareous substrata—are members of the Asteraceae (82 spp.), Fabaceae (46 spp.), Aizoaceae (41 spp.), Rutaceae (41 spp.) and Iridaceae (27 spp.) (Fig. 7A; Table S2). It is noteworthy that the Scrophulariaceae, Poaceae and Cyperaceae, three cosmopolitan families, comprise several calcicoles (24, 18 and 10 spp., respectively), as do typical Cape families like the Ericaceae (24 spp.), Proteaceae (11 spp.) and Restionaceae (16 spp.). Interestingly among calcicolous graminoids, a third of calcicolous grasses belong to the Cape genus *Pentameris* (6 spp.), while most calcicolous sedges belong to the genus *Ficinia* (6 spp.), a typical Southern African lineage.

While there are similarities in the familial composition between the calcarenite-endemic and the dune-endemic portions of the coastal flora (Figs. 7C, 7D; Table S3), for example the dominance of the Asteraceae and prominence of the geophytic Iridaceae, there are also important differences. Cape families, such as the Ericaceae, Fabaceae, Restionaceae and Rutaceae, are more pronounced in the calcarenite-endemic flora, while the Proteaceae, which is rich in calcarenite-endemics, is, apart from sporadic occurrences of *Leucadendron coniferum* on dunes along the Cape Peninsula and the western Agulhas Plain coast, absent from the dune flora. On the other hand, cosmopolitan families like the Asteraceae and Poaceae, the pantropical Celastraceae, and the largely southern-African succulent families Aizoaceae and Asphodelaceae, are more prominent in the dune-endemic flora. Additionally, the Amaranthaceae and Plumbaginaceae (*Limonium*), families with several salt-tolerant members, occur in the dune-endemic flora (3 and 4 spp., respectively), but contribute no calcarenite-endemic species (though four calcicolous *Limonium* species also occur on calcarenites).

All calcicolous Ericaceae belong to *Erica* (24 spp.)—the genus contributing the greatest number of calcicoles to the coastal flora—with most species being endemic to calcarenites (19 spp.). The same is true for *Aspalathus* (12 of 16 spp. are calcarenite endemics), *Muraltia* (seven of 12 spp.), *Phylica* (six of nine spp.), *Diosma* (five of six spp.) and *Euchaetis* (four of six spp.). In other calcicole-rich genera, the split between calcarenite- and dune-endemic species is more equal, for example *Indigofera* (six and four spp., respectively of 12 spp. total) and *Hermannia* (four and four spp., respectively of 10 spp. total). In *Agathosma*—the second richest genus in terms of calcicoles—most species are

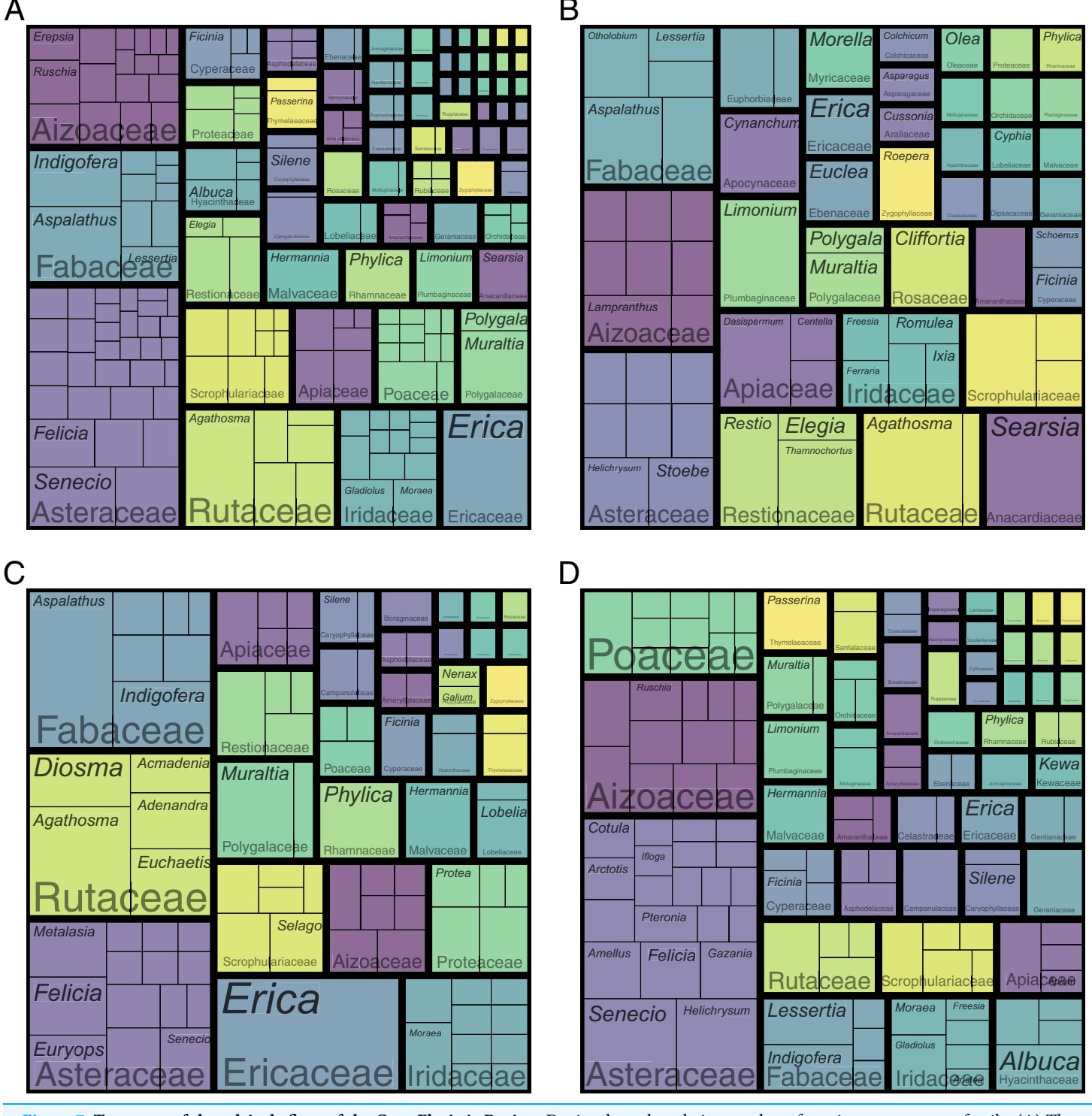

**Figure 7 Treemaps of the calcicole flora of the Cape Floristic Region.** Depicted are the relative number of species per genus per family. (A) The total calcicole flora (species endemic to calcareous substrata; 548 spp.). (B) The calcicole-ubiquist flora (calcicolous species occurring on both dunes and calcarenites; 104 spp.). (C) The calcarenite-endemic flora (218 spp.). (D) The dune-endemic flora (226 spp.). Note that B–D are subsets of A. Note that colours only aid to differentiate different plant families.

restricted to calcarenites (10 of 19 spp.), although several are dune-endemics (four spp.) or calicole ubiquists (five spp.). *Searsia*, a genus with subtropical affinities, contributes seven calcicoles to the flora, with most species (five spp.) occurring on both calcarenites and dunes. Dune endemism is pronounced in the calcicole-rich asteraceous genera *Helichrysum* (seven of nine spp. are dune endemics) and *Senecio* (eight of 12 spp.), while the same is true for *Wahlenbergia* (four of six spp.) and the $C_3$-grass genus *Pentameris* (four of six spp.).

Compared to the whole coastal flora, the calcicolous component shows a stronger biogeographic affinity to the Greater Cape, with nearly half (49%, 267 spp.) of calcicoles belonging to genera typical of the GCFR. This is pronounced in the calcarenite-endemic flora (examples shown in Fig. 8), where 64% (139 spp.) belong to GCFR-genera, while the Greater-Cape affinity in the dune-endemic flora (35%, 78 spp.) (examples shown in Fig. 9) is the same as that of the whole coastal flora. Species belonging to tropical genera comprise 8% (43 spp.) of calcicoles—about half the proportion in the rest of the coastal flora. The affinity to tropical floras is evident among dune-endemics (11%, 25 spp.), but muted among calcarenite-endemics (<3%, 6 spp.).

Calcicolous species in the Cape coastal flora are overwhelmingly GCFR-endemics (92%, 506 spp.), with a large majority of these species being restricted to the CFR (84% of calcicoles, 461 spp.). Calcicolous species with distributions that extend beyond the GCFR are largely associated with dunes (33 spp.), with a smaller number occurring on both dunes and calcarenites (nine spp.), though none are calcarenite endemics. More than a third (35%, 194 spp.) of calcicolous species are endemic to the Agulhas Plain centre (AP), most of them being calcarenite-endemic species (164 spp.) and only a few being dune endemics (14 spp.). The Southeastern (SE) and Southwestern centres (SW) each have about 11% (each 59 spp.) of calcicoles endemic to them. Dune endemism (44 spp.) is pronounced among SE-endemic calcicoles, while the number of calcarenite endemics (four spp.) is low. Among SW-endemic calcicoles, numbers of species restricted to calcarenites (21 spp.) and dunes (29 spp.) are comparable. Levels of local-endemism are highest among AP-endemic calcicoles (112 spp.), followed by SW- (44 spp.) and SE-endemic calcicoles (22 spp.).

Most calcicoles are dwarf shrubs (35%, 193 spp.) or low shrubs (19%, 104 spp.), both among calcarenite- and dune-endemic species. Forbs (12%, 68 spp.) and geophytes (11%, 59 spp.) are common in the calcicolous flora, with the former being primarily dune endemics (48 spp.) and the latter split more equally between calcarenite- (22 spp.) and dune-endemic species (29 spp.). Of the 34 calcicolous annuals found in the CFR coastal flora, 22 are restricted to dunes.

# DISCUSSION

## Comparison with other Cape floras

The flora of the Cape is highly distinctive, evident through its recognition as a distinct biogeographic area for nearly two-and-a-half centuries (*Bolus, 1886*; *Goldblatt, 1978*; *White, 1983*; *Takhtajan, 1986*; *Linder et al., 2005*; *Born, Linder & Desmet, 2007*; *Colville et al., 2014*). The evolution of this flora is intimately tied to the Cape Fold Belt, an ancient mountain range mainly composed of Ordovician–Devonian quarzitic sandstones that

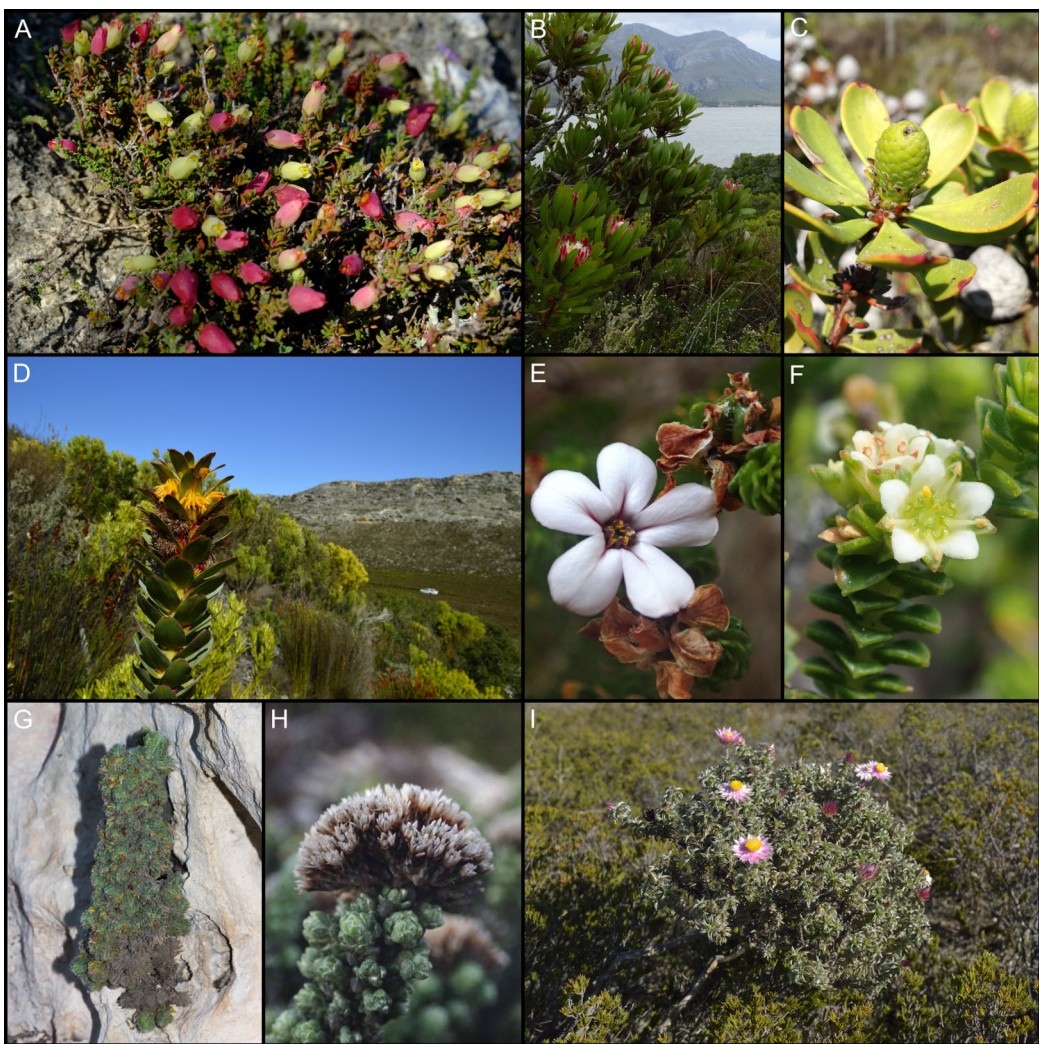

**Figure 8** **Examples of calcarenite-endemic species from the coastal flora of the Cape Floristic Region.**
(A) *Brachysiphon mundii* of the Penaeaceae. (B) *Protea obtusifolia* of the Proteaceae. (C) *Leucadendron muirii* of the Proteaceae. (D) *Mimetes saxatilis* of the Proteaceae. (E) *Adenandra obtusata* of the Rutaceae.
(F) *Diosma guthriei* of the Rutaceae. (G) *Erica occulta* of the Ericaceae. (H) *Metalasia calcicola* of the Asteraceae. (I) *Achyranthemum recurvatum* of the Asteraceae. Most calcarenite endemics belong to Cape lineages. Image credits: (A) Nick Helm, inaturalist.org 22021, licensed under CC BY 3.0 SA; (B–F, H, I)
B. Adriaan Grobler; (G) Ross C. Turner.               

weather to shallow, acidic, nutrient-poor, sandy soils. The diversification of several Cape
lineages is closely linked to these soils (*Hoffmann, Verboom & Cotterill, 2015*; *Van Santen
& Linder, 2019*), which have covered extensive areas of the Cape since the sandstones
were exhumed following post-Gondwanan, Late-Cretaceous–early-Cenozoic erosion
(*Partridge, 1998*; *Tinker, De Wit & Brown, 2008*). Sea-level fluctuations during the
Plio–Pleistocene saw, for the first time in the evolutionary history of the CFR, the
deposition and exposure of large tracts of calcareous substrata along the coast, providing
novel edaphic environments for the colonization and ensuing diversification of the
Cape flora (*Cowling, Proches & Partridge, 2009*). How distinct is this coastal flora—the

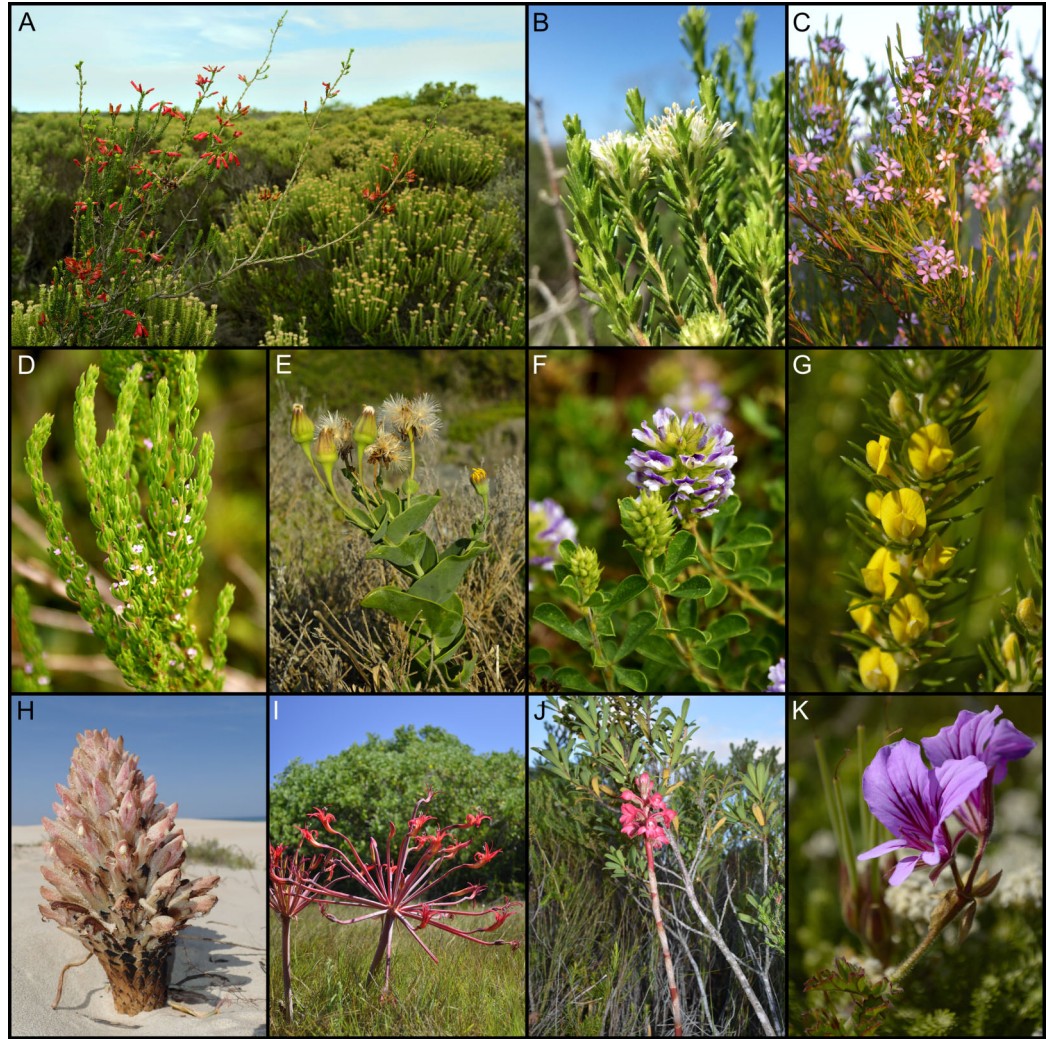

**Figure 9 Examples of dune-endemic species from the coastal flora of the Cape Floristic Region.**
(A) *Erica chloroloma* of the Ericaceae. (B) *Agathosma stenopetala* of the Rutaceae. (C) *Coleonema pulchellum* of the Rutaceae. (D) *Muraltia satureioides* of the Polygalaceae. (E) *Othonna rufibarbis* of the Asteraceae. (F) *Otholobium* sp. nov. 'algoensis' of the Fabaceae. (G) *Aspalathus cliffortiifolia* of the Fabaceae. (H) *Hyobanche robusta* of the Orobanchaceae. (I) *Brunsvigia litoralis* of the Amaryllidaceae. *Satyrium princeps* of the Orchidaceae. (K) *Pelargonium suburbanum* subsp. *suburbanum* of the Geraniaceae. Most dune endemics belong to Cape (A–G) or Southern African (H–K) lineages. Image credits: B. Adriaan Grobler.                             

product of the most recent colonization and diversification event in the CFR—from the Cape's typically calcifuge (intolerant of alkaline soils) flora, and what is its contribution to the globally unique plant diversity of this megadiverse region?

The coastal flora, comprising 1,365 species, accounts for nearly 15% of flowering-plant species in the CFR (ca. 9,300 spp. in total) (*Manning & Goldblatt, 2012a*). While our results showed that most of these species are edaphically widespread, a significant portion (548 spp.) of the coastal flora is restricted to calcareous substrata. Thus, 6% of species in the Cape flora are strictly calcicoles with coastal distributions. Both the contemporary ecological conditions and evolutionary history of CFR coastal lowlands differ from that of

the inland mountainous regions that gave rise to the ancestral Cape flora. These differences are most pronounced in dune habitats, where a unique selective regime has resulted in regional-scale floras that are typically poorer in species compared to inland floras, albeit marginally (*Grobler et al., 2020*). Central to this selective regime was the vacillating sea levels of the Pleistocene that led to repeated drowning and exposure of coastal dunes. This instability likely induced high extinction rates in the limited pool of species that were able to colonize the harsh dune environment; here, species must overcome challenges imposed on them by high solar radiation and strong, salt-laden winds throughout the year, the latter of which also carry highly abrasive sand grains and can bury or excavate plants through sand movement (*Wilson & Sykes, 1999*; *McLachlan & Brown, 2006*; *Illenberger & Burkinshaw, 2008*; *Maun, 2009*). In addition to these factors, plants growing in dunes are affected by the unique soil environment: dune sands are highly alkaline, have a poor water-holding capacity, and characterized by nutritional imbalances induced by the impact of high pH on nutrient uptake (*Brady, 1974*; *Maun, 2009*; *Pye & Tsoar, 2009*).

Calcarenite landscapes of CFR coastal lowlands house soils that are chemically and physically similar to those of dune landscapes, although they are generally far shallower and accumulate in fissures and potholes in the underlying calcarenites, thus presenting a challenging edaphic environment for plants. For example, the calcarenite endemics *Leucadendron meridianum* and *Protea obtusifolia* exhibit stunted growth (lower stature and smaller canopy volume) and reduced fecundity (fewer cones and fertile seeds) compared to their respective sister taxa, *Leucadendron coniferum* and *Protea susanae*, that grow in adjacent, deep colluvial sands on the Agulhas Plain (*Mustart & Cowling, 1993*; *Mustart, Cowling & Dunne, 1994*). For species that typically grow in deep dune sands, stunted growth appears to be even more pronounced in individuals that have managed to colonize small outcrops of coastal calcarenites: in *Erica glumiflora*—a dune endemic of the southeastern CFR—typical height in its native dune habitat is ca. 0.6–1.0 m (*Schumann, Kirsten & Oliver, 1992*; *Oliver, 2012*), whereas plants on calcarenite rarely grow more than 0.1 m tall (Fig. 10). While calcarenite outcrops do occur near the shore, they are most extensive inland of coastal dunes, and plants associated with this substrate are therefore less affected by the typically coastal disturbances active in dune environments (*e.g.*, strong winds, salt spray, sand movement). Furthermore, calcarenite landscapes that occur inland of the current coastline have not been directly influenced by sea-level changes since the highstand sequences of the late Pliocene (*Partridge & Maud, 2000*; *Cowling, Procheş & Partridge, 2009*). This means that, both presently and historically, calcarenite landscapes in the CFR present more stable environments than do coastal dunes, and selection pressures exerted here are likely overwhelmingly edaphic in nature.

The selective forces active in coastal calcareous landscapes of the Cape have produced a distinctive floral assemblage comprising 226 dune-endemic species and 218 calcarenite-endemic species (each group comprising nearly 3% of the entire CFR flora), with a further 104 species restricted to coastal calcareous substrata more generally—a high tally given the comparatively short but turbulent past experienced by coastal dune and calcarenite landscapes in the Cape.

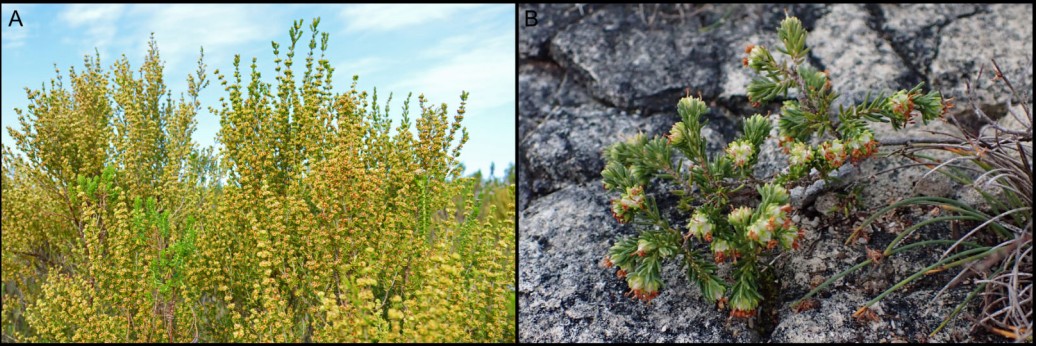

**Figure 10 Growth on different coastal calcareous substrata of the dune-endemic *Erica glumiflora* (Ericaceae) from the southeastern Cape Floristic Region.** (A) Typical plant stature, reaching heights of 0.6–1.0 m, in deep dune sand. (B) Atypical stunted growth on calcarenite, with plants rarely growing taller than 0.1 m. Photographs by B. Adriaan Grobler.

Not unexpectedly, there are similarities in the floristic composition between the CFR coastal flora and the calcifuge floras associated with montane, acidic soils (*McDonald & Morley, 1988*; *Taylor, 1996*; *McDonald, 1999*). Note that we include in this acidophile profile floras associated with windblown, acid sands on the lowlands, since these substrata support floras more similar to montane floras than any other soil group in the CFR (*Cowling et al., 1988*; *Thwaites & Cowling, 1988*). At the family level, the identity and rank of the three largest plant families—the Asteraceae, Fabaceae and Iridaceae—are the same in both floras, while the Aizoaceae and Scrophulariaceae also occupy similar ranks. There are, however, also striking differences, even at this high taxonomic level: the high numbers of species of Ericaceae, Proteaceae and Restionaceae characteristic of montane floras are not mirrored in the coastal flora, with none of these families featuring in the ten richest families of the latter flora; instead, the Rutaceae and the two graminoid families Cyperaceae and Poaceae feature more prominently in the coastal flora, while the Apiaceae is also more strongly represented. Of the endemic families (Bruniaceae Geissolomataceae, Grubbiaceae, Penaeaceae, Roridulaceae) of the Cape flora (*Goldblatt, 1978*; *Manning & Goldblatt, 2012a*), only the Bruniaceae and Penaeaceae are represented in the coastal flora, though each by only a single species: *Brunia laevis* (Bruniaceae), occurring on calcarenite and sandstone substrata, and *Brachysiphon mundii* (Penaeaceae), a range-restricted calcarenite endemic of the Agulhas Plain Centre. It should be noted that, of the Cape-endemic families, the Bruniaceae (six genera, 78 spp.) and Penaeaceae (seven genera, 23 spp.) are by far the most diverse (all others are monogeneric and mono- or oligotypic) (*Manning & Goldblatt, 2012a*), and it is not surprising that these two families have managed to colonize and speciate (in the case of Penaeaceae) on calcareous substrata, even though it is only to a limited extent.

At the generic level, there are again similarities between the CFR coastal flora and montane floras, although they are limited: in both cases, *Erica* and *Aspalathus* are the most speciose genera, while *Agathosma* ranks fourth and third in these floras, respectively. *Restio*, a high-ranking genus in montane floras, does not feature in the ten largest genera of the coastal flora, while the same is true for other speciose genera in montane floras,

including *Phylica*, *Cliffortia* and *Oxalis*. Genera that feature more prominently in the coastal flora in comparison to montane floras are *Ficinia*, *Hermannia*, *Indigofera*, *Crassula* and especially the asteraceous genera *Senecio* and *Helichrysum*. Thus, while some Cape-centred genera are well-represented in the coastal flora, others are not, instead being replaced by genera that have their centres of diversity outside of the CFR.

The ratio of species per genus (S/G-ratio) in the coastal flora (3.1) is comparable to that of regional floras from the southeastern CFR (mean S/G-ratio = 3.1 ± 0.55) (*Cowling & Holmes, 1992a*), but slightly lower than southwestern floras (3.7 ± 0.47) (*Cowling & Holmes, 1992a*), which tend to be richer in species (*Cowling, Holmes & Rebelo, 1992*; *Cowling & Lombard, 2002*). The moderate species-to-genus ratio of the coastal flora suggests that there has not been extensive radiation of lineages, although the Rutaceae is one important exception, contributing eight genera and 61 species to the coastal flora (S/G-ratio of 6.8), of which 41 species are endemic to calcareous substrata. Nearly half of the calcicolous Rutaceae belong to *Agathosma* (19 spp.), a genus which has also diversified extensively on acidic substrata of the CFR (*Manning & Goldblatt, 2012a*). While subdued in comparison with floras of the Cape hinterland, there has also been some diversification among other species-rich genera in the coastal flora, for example *Ficinia*, *Helichrysum*, *Hermannia*, *Indigofera*, *Senecio*, *Phylica*, *Thamnochortus*, and especially the Cape-centred genera *Aspalathus*, *Erica* and *Muraltia*. Unique among tropical genera is *Searsia*, which has also diversified since colonization of calcareous substrata.

Growth-form composition of the coastal flora mirrors some characteristics of montane floras, for example the sparsity of trees, which comprise less than 3% of species in both cases, and the dominance of shrubs, which constitute about half of all species in both coastal and montane floras (*Goldblatt, 1978*; *Manning & Goldblatt, 2012a*). While shrubs of lower stature, especially dwarf shrubs, are most common in the coastal flora, it is unclear what proportion of montane floras they constitute. Given the selective pressures active in the coastal flora, specifically the persistence of strong winds, it seems likely that a lower stature would impart a selective advantage, and that low growth forms like dwarf shrubs would be more frequent in the coastal flora than in inland floras of the Cape. A peculiarity of the whole Cape flora, particularly among MCEs, is the exceptionally high proportion of geophytic species (*Proches et al., 2006*), comprising nearly 20% of the total flora (*Goldblatt, 1978*; *Manning & Goldblatt, 2012a*) and comprising more than 10% of regional montane floras (*e.g.*, *McDonald, 1999*). Our coastal flora has a slightly higher tally (15% of species are geophytes), while the major families that contribute to this diversity—the Iridaceae, Hyacinthaceae, Orchidaceae, Amaryllidaceae, Asphodelaceae and Oxalidaceae—are the same and maintain similar rankings. Although the number of annual species in the Cape is not low, they are proportionally underrepresented in comparison with other growth forms and with other MCEs, making up only 6.5% of the Cape flora as a whole (*Manning & Goldblatt, 2012a*) and having even lower numbers in montane floras (*Taylor, 1978*). The proportion of annuals in the coastal flora is higher at 9%, although still relatively low compared to regions of similar climate like California or Chile (*Cowling et al., 1996*). As in the montane flora of the southern Langeberg (*McDonald, 1999*), most annual species in the coastal flora belong to the Asteraceae and

Scrophulariaceae, and while the Gentianaceae and Lobeliaceae also contribute a large proportion of annuals to the acidophilous Langeberg flora, these families are poorly represented in the coastal annual flora.

Although the bulk of the Cape flora comprises species associated with the geologically ancient Cape Fold Belt, there are subsets of species that are intimately associated with more recently exhumed substrata on the CFR lowlands. Similar to the flora of coastal calcareous substrata, floras of renosterveld—a vegetation type associated with shale geologies and alluvial deposits that give rise to nutrient-rich loamy soils (*Boucher & Moll, 1981*)—are a comparatively recent assemblage that emerged during the Mio–Pliocene (*Cowling, Procheş & Partridge, 2009*; *Hoffmann, Verboom & Cotterill, 2015*). Unfortunately, no comprehensive renosterveld flora exists, but we can gain some insight into the characteristics of renosterveld floras from the limited and disparate site-scale inventories that exist in the literature (*Walton, 2006*; *Kraaij, 2011*; *Curtis, 2013*; *Cowan & Anderson, 2014*). In these floras, the Asteraceae, Fabaceae, Poaceae and Iridaceae are typically the dominant families, while the Aizoaceae, Asphodelaceae, Cyperaceae, Hyacinthaceae, Oxalidaceae and Scrophulariaceae are also speciose (Table S4; *Kraaij, 2011*). Other families that often rank among the twenty most species-rich families include the Amaryllidaceae, Crassulaceae, Geraniaceae, Hypoxidaceae, Polygalaceae and Thymelaeaceae. The most speciose genera are typically *Aspalathus*, *Crassula*, *Helichrysum*, *Moraea*, *Oxalis* and *Pelargonium*, while *Hermannia*, *Ornithogalum* and *Senecio* feature prominently in some floras. Notable is that other geophytic genera, such as *Babiana*, *Drimia*, *Lachenalia*, *Romulea*, *Spiloxene* and *Trachyandra*, commonly harbour several species. Although they are generally not rich in species, genera of tropical affinity, especially *Asparagus*, *Euclea*, *Olea* and *Searsia*, are common components of renosterveld floras.

How do renosterveld floras compare with the coastal flora of the CFR? The dominant families—Asteraceae, Fabaceae, and Iridaceae—are the same in renosterveld and the coastal flora, but the graminoid families Cyperaceae and Poaceae rank higher in renosterveld, whereas the Rutaceae—a high-ranking taxon in the coastal flora—forms only a minor component in renosterveld floras. Among genera, typically shrubby lineages like *Aspalathus*, *Helichrysum*, *Hermannia* and *Senecio* occupy similarly high ranks in the coastal flora and renosterveld floras. *Crassula* and *Pelargonium*, as well as geophytic lineages, especially *Moraea* and *Oxalis*, rank higher in renosterveld. As renosterveld harbours a disproportionately high diversity of geophytes in the Cape (*Cowling, 1990*; *Procheş et al., 2006*), it is not surprising that typically geophytic families (Amaryllidaceae, Hypoxidaceae, Orchidaceae, Oxalidaceae) and genera (*Moraea*, *Ornithogalum*, *Oxalis*, *Spiloxene*) rank higher among speciose taxa in renosterveld floras compared to the Cape coastal flora. Another conspicuous difference between these floras regards the genera *Erica* and *Agathosma*: while they are the most and third-most speciose genera in the coastal flora, respectively, they are typically species-poor in renosterveld floras. Significant features shared between the coastal flora and renosterveld floras are the relatively low rank occupied by the Ericaceae, Proteaceae and Restionaceae—families that are species-rich in typical Cape fynbos floras (*Goldblatt, 1978*; *Cowling & Holmes, 1992a*; *Manning &*

*Goldblatt, 2012a*)—and the relatively high incidence of tropical lineages, especially prominent at the generic level.

In an analysis of dominant species in GCFR vegetation types, *Bergh et al. (2014)* showed that dune fynbos–thicket mosaics ('strandveld' in their terminology) shared floristic links with renosterveld, while limestone fynbos was floristically most similar to fynbos occurring on infertile acid sands. This suggests that the dune component of the Cape coastal flora resembles renosterveld floras more closely, whereas the calcarenite component mirrors the composition of calcifuge fynbos floras. Likely causes of the closer link between dune and renosterveld floras include the higher incidence of non-restioid graminoids (where the Cyperaceae and Poaceae occupy similar ranks as in renosterveld floras) and shrubs allied to subtropical thicket (*e.g.*, *Euclea*, *Olea*, *Searsia*) in dune floras (*Cowling et al., 2019*).

In terms of growth forms, renosterveld floras are typically dominated by geophytes and low-stature shrubs, followed by forbs (including several annual species) and graminoids (grasses and sedges), while trees and climbing species (lianas and vines) are rare (*Cowling, Pierce & Moll, 1986*; *Walton, 2006*; *Kraaij, 2011*; *Curtis, 2013*; *Cowan & Anderson, 2014*). Succulents, especially leaf-succulents (*Cowling, 1984*), are a common component of renosterveld floras, comprising in the order of 10% of species (*Walton, 2006*; *Kraaij, 2011*; *Curtis, 2013*). This growth-form profile is similar to that of the Cape coastal flora, though geophytes are a much more prominent feature of renosterveld floras, while dwarf shrubs, especially those of Cape affinity, constitute a larger portion of the coastal flora.

Generally, about a third of renosterveld species are endemic to the CFR, especially among shrubs, but geophytes and succulents also exhibit high levels of regional endemism, particularly in renosterveld of the western CFR (*Cowling, 1983*; *Cowling & Holmes, 1992a*). In the fynbos–renosterveld transitional vegetation of the Bontebok National Park, *Kraaij (2011)* found that 46% of species are CFR-endemics, while *Cowling (1983)* found a comparable level of regional endemism in a similar fynbos–renosterveld community in the southeastern Cape. These are modest levels of endemism compared to the coastal flora, in which nearly 60% of species are restricted to the CFR. More comparable is the regional endemism of Cape dune floras, where regional endemism ranges from ca. 30–40% (*Cowling et al., 2019*). No information is available on levels of edaphic endemism in renosterveld floras, although it is expected to be modest (*Cowling, 1983*), other than in habitats associated with locally unusual geologies and soils where edaphic endemism can be pronounced (*e.g.*, *Curtis, Stirton & Muasya, 2013*).

## Comparison with other calcareous-substrate floras

Information on directly comparable floras—coastal floras associated with calcareous substrata from other Mediterranean-climate ecosystems (MCEs)—is especially sparse, and to our knowledge, our study is the first to present a comprehensive analysis of a coastal flora from one of the world's hyperdiverse MCEs (cf. *Rundel et al., 2016*). The MCE-zone of central Chile supports well-developed coastal dune systems, of both Holocene and Pleistocene age, but coastal calcarenite formations appear to be a less important feature in this region (*Araya-Vergara, 2007*). From the limited investigations into coastal dune floras

of this region, it is apparent that they share with the Cape coastal flora a dominance by the Asteraceae, Fabaceae and Poaceae, with most species belonging to shrubby lineages (*San Martin, Ramírez & San Martin, 1992*). *Armesto, Arroyo & Hinojoa (2007)* suggest that several species occurring in Chilean coastal dune floras are dune endemics, but no data exists to support this claim.

In California, 'unusual' substrata typically comprise serpentine soils, and floras associated with these ultramafic substrata have been the main focus of research into edaphic adaptation and endemism in this region (*e.g.*, *Kruckeberg, 1984*). The flora of the White Mountains, a range comprising extensive dolomite (calcium–magnesium carbonate rock) outcrops, has been well studied (*e.g.*, *Lloyd & Mitchell, 1973*; *Rundel, Gibson & Sharifi, 2008*), but it occurs in the Desert and Alpine biomes of eastern California, away from the coast and outside of the Mediterranean-climate zone. Calcareous dunes and calcarenites do occur along the southern Californian coast (*Cooper, 1967*), but studies into their specific floras are limited, focusing largely on geologically young (Holocene) coastal dune landscapes (*e.g.*, *Purer, 1936*; *Williams & Potter, 1972*; *Johnson, 1977*; *Barbour et al., 1981*; *Pickart & Barbour, 2007*; *Peinado et al., 2011*; *US Fish & Wildlife Service, 2016*). As in the Cape coastal flora, these Californian floras are typically dominated by members of the Asteraceae, Fabaceae and Poaceae, although the Boraginaceae and Amaranthaceae also contribute a substantial proportion of species (*Purer, 1936*; *Barbour et al., 1981*; *Pickart & Barbour, 2007*; *US Fish & Wildlife Service, 2016*). A high proportion of these species are associated with the Coastal biome, but as in the CFR, most occur in Mediterranean-type shrublands, namely coastal chaparral (*Barbour et al., 1981*). Approximately 30–40% of species in Californian dune floras are restricted to the MCE-zone (*Johnson, 1977*; *Barbour et al., 1981*), a moderate level of regional endemism in comparison with the Cape coastal flora, although edaphic endemism in dunes of California, ranging from ca. 20–40% (*Johnson, 1977*; *Barbour et al., 1981*; *US Fish & Wildlife Service, 2016*), is comparable.

Calcareous substrata dominate most of the coastal and inland landscapes of the Mediterranean Basin (*Lewin & Woodward, 2009*), and we would therefore expect floras associated with Mediterranean coastal substrata to not have been subject to the ecological filter imposed by a strong alkalinity gradient, as is the case in the CFR. Nevertheless, as in the Cape, coastal dune floras of the Mediterranean Basin appear to be a regionally distinct formation, shaped by the strong selective pressures operating in these coastal habitats. Here, the Fabaceae, Asteraceae and Poaceae comprise nearly half of species in local floras, while members of the Amaranthaceae, Caryophyllaceae, Apiaceae and Plambaginaceae are also frequent (*Hadjichambis et al., 2004*; *Korakis & Gerasimidis, 2006*; *Ciccarelli, Di Bugno & Peruzzi, 2014*). On limestone formations, floras comprise several species of Fabaceae, Lamiaceae and Rosaceae, most of which are shrubby species associated with Mediterranean-type shrublands (*Kruckeberg, 2002*). Geographical endemism in limestone floras varies, but several local and regional endemics occur (*Kruckeberg, 2002*), while in coastal dune floras, about 70% of species are restricted to the Mediterranean Basin (*Hadjichambis et al., 2004*; *Spanou et al., 2006*; *Muñoz Vallés, Gallego Fernández & Dellafiore, 2009*; *Ciccarelli, Di Bugno & Peruzzi, 2014*; *Iliadou et al., 2014*)—a level similar

to that found in the Cape coastal flora. Edaphic endemism is believed to be high in Mediterranean calcareous floras (*Van Der Maarel & Van Der Maarel-Versluys, 1996*; *Kruckeberg, 2002*), but data to illustrate this are sparse. Dune endemics comprise ca. 18% of species in the Israeli coastal dune flora (*Barbour et al., 1981*; *Kutiel, 2001*).

Most akin to the Cape coast in terms of climate, geology and physiography are the dune and calcarenite landscapes of coastal Southwestern Australia, where some data are available on regional floras of the Swan Coastal Plain (*Dixon, 2011*; *Zemunik et al., 2016*) and the South Australian coast (Fleurieu Peninsula to Port Macdonnell) (*Oppermann, 1999*). Here, the Asteraceae, Fabaceae and Poaceae contribute most species to floras, while the Amaranthaceae, Cyperaceae and Proteaceae are also well represented (*Oppermann, 1999*; *Zemunik et al., 2016*)—a pattern similar to the Cape coastal flora. A feature that distinguishes Australian calcareous floras from that of the Cape is the high proportion of Myrtaceae (*Dixon, 2011*; *Zemunik et al., 2016*), a family represented in the CFR by a single Gondwanan relict species, *Metrosideros angustifolia*, but which is absent from calcareous substrata in the region, instead being restricted to riparian habitats of the Cape Fold Belt. Growth-form spectra of Australian coastal floras are also very similar to that of the CFR: they are dominated by shrubby species, the incidence of deciduous and evergreen hemicryptophytes is roughly equal, annuals are subsidiary, and, unusual among MCEs but similar to the Cape, the liana and vine flora is well developed (*Oppermann, 1999*; *Zemunik et al., 2016*). Levels of geographic endemism in these floras appear to be low compared to the Cape, for example *Cowling et al. (1994)* found no local endemics on calcareous substrata in 0.1 ha plots at the Barrens in Western Australia, while only 7% and 14% of species occurring on calcareous sand and calcarenites, respectively, were regional endemics.

*Morat, Jaffre & Veillon (1997)* provide a description of the flora associated with calcareous substrata on the tropical island archipelago of New Caledonia, recognized globally as a biodiversity hotspot (*Mittermeier et al., 2011*). Here, these substrata cover ca. 3,800 km$^2$ and support various habitats, including tropical rainforest, sclerophyll forest and coastal dunes. While the ecological settings and evolutionary histories of New Caledonia and the Cape are vastly different, the areas covered by calcareous substrata in these regions are similar, thus providing a basis for some rudimentary comparisons between richness and endemism in their floras (*Rosenzweig, 1995*). The New Caledonian calcareous flora comprises 488 plant species, 40% of which are endemic to the archipelago (*Morat, Jaffre & Veillon, 1997*). Only 87 species are calcicolous (*i.e.*, 18% edaphic endemism), with most species in the flora (82%) occurring on various other soil types, including ultramafic soils. Calcicoles comprise less than 3% of the regional flora (ca. 3,200 spp.) (*Morat, 1993*), and nearly half (42 spp.) of the calcicolous species are endemic to New Caledonia. The CFR coastal flora is thus far richer in species than the New Caledonian equivalent, with levels of regional and edaphic endemism also being higher in the Cape. Furthermore, among calcicolous species, regional endemism is much more pronounced in the Cape coastal flora, where calcicoles also comprise double the proportion of the regional flora compared to New Caledonia. The relative paucity of species and endemics on calcareous substrata of New Caledonia is not currently

understood, but *Morat, Jaffre & Veillon (1997)* suggest that prolonged degradation of calcareous habitats by human activity could partly explain this. As in the Cape, most of New Caledonia's calcareous substrata date from the Quaternary. It is thus likely that the calcicolous flora of New Caledonia emerged in the recent geological past, a result of immigrant species arriving *via* long-distance dispersal and their offspring diversifying during the Quaternary (*Morat, Jaffre & Veillon, 1997*). The relative youthfulness of this flora could therefore also be invoked to partially explain its depauperate nature with respect to the regional flora of New Caledonia, which started to emerge during the late Palaeogene and Neogene (*Pillon, 2012*), but not in comparison with the CFR coastal flora, which is of a similarly young age (*Cowling, Proches & Partridge, 2009*; *Hoffmann, Verboom & Cotterill, 2015*).

## Assembly of the Cape coastal flora

The biogeographic affinities of taxa in the CFR coastal flora points to an autochthonous assemblage that has largely been derived from the regional Cape flora; however, there has also been substantial historical input from desert, temperate and tropical floras from outside of the CFR. In this section, we relate the coastal flora to these abutting biogeographic areas that likely acted as the sources of its component species and sketch a brief scenario of its assembly.

Globally, the phylogenetic structuring of regional coastal floras indicate that their component species are typically derived from adjacent, inland floras, rather than being a product of long-distance dispersal from other geographically remote coastal areas (*Brunbjerg et al., 2014*). This is in accord with our analyses, which show that the Cape coastal flora is overwhelmingly a southern African assemblage, with nearly two-thirds of species belonging to genera with affinities to the subcontinent. Most of these—over a third of the coastal flora—are affiliated with the Greater Cape Floristic Region (GCFR), which includes the CFR as well as the winter-rainfall semi-deserts of the Namaqualand and Hantam–Tanqua–Roggeveld regions (*Born, Linder & Desmet, 2007*). The prominence of Cape clades (*sensu Linder, 2003*) like *Agathosma*, *Aspalathus*, *Diosma*, *Erica*, *Muraltia* and *Phylica* in the coastal flora, particularly its endemic (calcicolous) component, identifies it as a derivative of the globally distinct Cape flora, but a characteristic element of the typical calcifuge floras of the region that is lacking in the coastal flora is the marked diversity of the Restionaceae and Proteaceae (*Goldblatt, 1978*; *Cowling & Holmes, 1992a*; *Manning & Goldblatt, 2012a*). While these two families have given rise to several calcicolous species (16 Restionaceae, 11 Proteaceae), their diversification on calcareous substrata has been disproportionately limited in comparison with these families on acid sands of montane habitats in the CFR (cf. *Cowling & Lamont, 1998*; cf. *Linder, 2001*).

Other GCFR elements occurring in the Cape coastal flora are affiliated with the winter-rainfall semi-desert regions of western South Africa (cf. *Jürgens, 1997*; cf. *Cowling, Esler & Rundel, 1999*; cf. *Snijman, 2013*) and the Little Karoo (part of the CFR) (cf. *Vlok & Schutte-Vlok, 2015*) where the dominant vegetation formation is the Succulent Karoo biome. These overwhelmingly comprise low or succulent dwarf shrubs, best represented in

the coastal flora by the Aizoaceae (*Carpobrotus*, *Drosanthemum*, *Erepsia*, *Lampranthus*) and Asteraceae (*Othonna*), while several geophytic taxa occurring in the coastal flora, for example Hyacinthaceae (*Albuca*, *Lachenalia*, *Massonia*), Iridaceae (*Babiana*, *Moraea*, *Romulea*) and Oxalidaceae (*Oxalis*), are also typical components of semi-desert floras in the GCFR.

Various arid-adapted taxa (families and genera) straddle the divide between the winter-rainfall GCFR and the summer-rainfall semi-deserts (Nama Karoo biome) of southern Africa (*Jürgens, 1997*; *Cowling & Hilton-Taylor, 1999*)—typically, these are species of southern-African, but extra-Cape, affinity. In the CFR coastal flora, such species include several succulents, especially among the Aizoaceae (*Delosperma*, *Galenia*, *Mesembryanthemum*, *Ruschia*), Asphodelaceae (*Aloe*, *Bulbine*), Asteraceae (*Crassothonna*) and Crassulaceae (*Crassula*, *Cotyledon*). The Asteraceae further contributes several shrubby, arid-adapted, southern-African lineages to the coastal flora, including *Chrysocoma*, *Felicia*, *Eriocephalus*, *Oncosiphon* and *Pteronia*, as do the Fabaceae (*Lessertia*, *Melolobium*), Scrophulariaceae (*Jamesbrittenia*, *Selago*) and Zygophyllaceae (*Roepera*). Typical of arid southern African floras, *Hermannia* (Malvaceae) is among the most diverse shrubby genera in the coastal flora, although nearly half (10 spp.) of the component species are coastal calcicoles (most of these belong to subgenus *Hermannia*, which is centred in the CFR) (*Verdoorn, 1980*). Several annuals—a growth form frequently associated with desert climates (*Van Rooyen, 1999*; *Klak & Bruyns, 2012*)—occurring in the coastal flora belong to higher taxa typical of southern African arid floras, with the Asteraceae (*Cotula*, *Helichrysum*, *Senecio*) and Scrophulariaceae (*Zaluzianskya*) especially well represented.

As the coastal habitats of the western CFR were directly linked with those of South Africa's arid west coast throughout the Plio–Pleistocene, there would have been few barriers impeding the southward migration of these winter-rainfall desert elements into the Cape during periods of lower rainfall. Other desert elements affiliated with the interior of southern Africa likely colonized coastal areas of the Cape during periods of similarly arid climates, although these interior source floras did not have a coastal distribution during the Plio–Pleistocene; rather, these elements probably migrated coastward *via* major river valleys that drain the arid interior, such as the Gouritz, Groot/Gamtoos, Sundays and Fish (*Cowling, 1983*).

During Pleistocene glacials, coastal forelands of the CFR and Palaeo-Agulhas Plain (PAP) were dominated by fire-prone vegetation, including fynbos, grasslands, renosterveld and alluvial woodlands, while fire-sensitive subtropical thicket was largely restricted to refugial river valleys (*Cowling et al., 2020*) following a marked regional contraction of this biome during the Neogene (*Vlok, Euston-Brown & Cowling, 2003*; *Cowling, Proches & Vlok, 2005*; *Potts et al., 2013*; *Neumann & Bamford, 2015*). The complex microtopography of coastal dunes, which were widespread on the PAP at this time (*Cawthra et al., 2020*), would have provided fire-sheltered sites in which dune thicket could persist (cf. *Cowling et al., 1997a*; cf. *Cowling & Potts, 2015*), despite the presence of fire in these systems at potentially higher frequencies than present (*Kraaij et al., 2020*). Contemporaneously with the establishment of these expansive dune areas, the summer-rainfall zone over
southern Africa extended further west into the CFR compared to the contemporary climate zones, particularly during glacials of the mid to late Pleistocene (*Engelbrecht et al., 2019*), thus allowing these calcareous substrata to act as a corridor for tropical lineages to migrate westward along the coast from the subtropical eastern seaboard of southern Africa into the Cape (*Cowling, 1983*). Reductions in rainfall and temperature during these glacial periods was remarkably tempered along the coast (*Engelbrecht et al., 2019*) and would therefore not have posed a major obstacle to the expansion of tropical lineages. It is also important to note that, prior to the establishment of the winter-rainfall regime over southwestern Africa and the expansion of fire-prone vegetation during the Neogene, subtropical thickets and forests dominated the area now recognized as the CFR (*Neumann & Bamford, 2015*). Thus, subtropical floras likely have a long history in the region, exemplified by the presence of putative palaeoendemic genera like *Heeria* (Anacardiaceae), *Hartogiella* and *Maurocenia* (Celastraceae), *Lachnostylis* (Phyllanthaceae), *Hyaenanche* (Picrodendraceae) and *Smelophyllum* (Sapindaceae) in the Cape flora (*Manning & Goldblatt, 2012a*). Recruitment of tropical lineages into the Cape coastal flora was therefore not necessarily limited to species migrating westward along the coastal margin, but likely involved their incorporation directly from those elements that have persisted in the region since the Palaeogene (*Cowling, Procheş & Vlok, 2005*). This is reflected in the prominence of tropical lineages in the Cape coastal flora, both those with Afrotropical and pantropical affinities. Interestingly, none of the ancient Cretaceous lineages characteristic of the southern African thicket flora, for example *Encephalartos* and *Strelitzia* (*Cowling, Procheş & Vlok, 2005*), have colonized calcareous substrata in the Cape, and are therefore absent from our flora. However, several basal lineages that evolved during the Eocene, such as the Celastraceae and Sapindaceae (*Cowling, Procheş & Vlok, 2005*), are characteristic components of dune thicket communities; in the case of the former, there has even been a marked diversification on coastal calcareous substrata, precipitating the evolution of four CFR-endemic calcicoles (*Cassine peragua* subsp. *barbara*, *Maurocenia frangula*, *Maytenus lucida*, *Robsonodendron maritimum*).

## Speciation in the Cape coastal flora

Most species in the Cape coastal flora have been directly incorporated from adjacent floras associated with older landscapes of the CFR, although a substantial portion have speciated on geologically young, calcareous substrata. A growing body of work (*Verboom, Linder & Stock, 2004*; *Malgas et al., 2010*; *Schnitzler et al., 2011*; *Hoffmann, Verboom & Cotterill, 2015*; *Verboom, Stock & Cramer, 2017*; *Van Santen & Linder, 2019*) has provided support for the hypothesis that the edaphic complexity of the CFR has acted as an important catalyst for ecological speciation in the Cape flora by creating a mosaic of divergent selection pressures (*Goldblatt, 1978*; *Cowling, 1987*; *Linder, 2003*). This hypothesis predicts that closely related taxa with sympatric distributions should occur on juxtaposed edaphic substrata, and such patterns have been demonstrated for various clades and soil types. In his revision of the genus, *Grau (1973)* notes that several *Felicia* species with broadly sympatric, coastal distributions in the Cape, including the calcicoles *F. amelloides*, *F. amoena* subsp. *latifolia* and *F. echinata*, appear to not be sister taxa and

are thus likely the product of ecological speciation. Indeed, it seems that most calcicolous species in the coastal flora have their sister taxa on non-calcareous substrata, suggesting that *in-situ* diversification of dune and calcarenite endemics has generally been limited, and that ecological speciation has been the major mechanism for diversification in the Cape coastal flora. This is consistent with the moderate species-to-genus ratio we report for the coastal flora.

*Rourke (1998)* describes evolutionary lines, based on changes in morphology, for the two proteaceous genera *Leucospermum* and *Spatalla*, in which species with primitive characters occur in geologically older montane habitats (*i.e.*, sandstone substrata), while more derived, specialized species are associated with geologically young (Plio-Pleistocene) deposits on coastal lowlands, especially along the Agulhas Plain. In *Spatalla*, the primitive and derived species form two distinct clades, with section *Cyrtostigma* comprising the older montane species and section *Spatalla* comprising the younger, mostly lowland species. While most species of the lowland clade are associated with siliceous substrata, one species, *Spatalla ericoides*, has evolved on calcarenites, and the general evolutionary pattern demonstrates the influence of Plio-Pleistocene geomorphic evolution along the Cape coast on speciation events and diversity patterns in the CFR.

Is there further evidence of ancient sandstone substrata in the Cape acting as the source of calcicole evolution? In the case of *Leucospermum*, *Rourke (1972)* first highlighted the importance of edaphic speciation by showing that various pairs of sister taxa were split between alkaline, calcarenite-derived soils and acidic, sandstone-derived soils. Other examples of this acid–alkaline edaphic divide have also been documented for sister taxa in other Cape lineages, including *Aspalathus* (*Dahlgren, 1963*, *1968*), *Freesia* (*Goldblatt, 1982*; *Manning & Goldblatt, 2010*), *Leucadendron* (*Williams, 1972*; *Barker et al., 2004*), *Muraltia* (*Levyns, 1954*; *Forest et al., 2007*), the *Pentaschistis* clade of *Pentameris* (*Linder & Ellis, 1990*; *Galley & Linder, 2007*), *Protea* (*Rourke, 1980*; *Schnitzler et al., 2011*) and *Thamnochortus* (*Linder & Mann, 1998*). Indeed, in cases where information is available, it seems that most calcicoles, especially those of Cape affinity, have their sister taxa on acid sands. This pattern is consistent with our analysis of edaphic associations in the coastal flora, which showed that most species with some edaphic affinity (*i.e.*, excluding species occurring on all soil types) were shared between calcareous coastal sands, windblown acid sands and sandstone-derived acid sands, suggesting that floras associated with these siliceous substrata have been the most significant source for colonization of calcareous substrata in the Cape. Very few species occurring in the coastal flora are shared exclusively between calcareous substrata and neutral loams—the typical substrate of renosterveld—suggesting that renosterveld floras have largely been excluded as a source of colonizing species in the coastal flora.

*Achyranthemum*, a genus of asteraceous dwarf shrubs recently segregated from *Syncarpha* (*Bergh & Manning, 2019*), provides an interesting example of late-Pliocene diversification (*Bergh, Haiden & Verboom, 2015*) resulting in a lineage comprising mostly calcicolous species (4 of 7 spp.). Furthermore, and unique among Cape lineages, the genus is centred in the coastal lowlands of the southeastern CFR. Molecular phylogenetic analysis places the dune endemic *Achyranthemum sordescens* as sister to the more

widespread *Achyranthemum striata*, a species typically occurring on lowland acid sands, whereas the closest relative of *Achyranthemum mucronatum*, a calcarenite endemic, is the widespread *Achyranthemum paniculatum*, which is most abundant on sandstone-derived acid sands (*Bergh & Manning, 2019*). While not included in the molecular analysis, morphological characteristics suggest that *Achyranthemum argenteum*, a dune-endemic species, is closely related to *Achyranthemum affine*, a species typical of lowland sandstone outcrops. The most restricted species in the genus, *Achyranthemum recurvatum*, is confined to calcarenites around Algoa Bay and appears to be phylogenetically isolated, being sister to the *Achyranthemum-chlorochrysum–mucronatum–paniculatum* clade (*Bergh & Manning, 2019*). As with most Cape lineages discussed thus far, the diversity patterns within *Achyranthemum* are most consistent with speciation driven by the ecological opportunities presented by novel edaphic substrata (*Bergh, Haiden & Verboom, 2015*), specifically diversification following colonization of coastal calcareous substrata from an inland, acid-sand source.

While the bulk of ecological speciation in the coastal flora has been among Cape lineages, there are also examples of tropical lineages colonizing and speciating on calcareous substrata of the CFR. In these cases, the calcicolous species are typically sister to edaphically widespread species that occur in inland thickets and forests. *Rapanea gilliana*, a low shrub endemic to dunes of the St Francis Bay–Algoa Bay area, is closely related to the tree *R. melanophloeos*, which occurs in forests throughout southern Africa (*Cowling, 1983*). A widespread calcicolous shrub occurring along the entire CFR coast, *Olea exasperata*, is a close relative of *Olea capensis*, typically a tall tree found in forests of southern and tropical Africa (*Besnard et al., 2009*). Another widespread dune-endemic shrub from the eastern CFR, *Maytenus procumbens*, is sister to *Maytenus undata*, a tree species typically growing in inland thicket and forests (*Simmons et al., 2008*; *McKenna et al., 2011*), while *Maytenus lucida*, a dune-endemic shrub restricted to the western CFR, is apparently derived from *M. procumbens* (*Cowling, 1983*)—this may be an example of limited calcicolous cladogenesis in this pantropical genus, though further phylogenetic study of this apparent lineage is required to elucidate relationships among its members.

Is there any evidence for *in-situ* diversification among Cape lineages in the coastal flora? While not supported by any molecular studies, *Dahlgren (1988)* proposes close affinities between some calcicolous *Aspalathus* species. Within his 'Adnates' group, for example, *Aspalathus pallescens*, *Aspalathus prostrata* and *Aspalathus salteri*—all calcarenite endemics—are the closest relatives to each other. He also recognizes a 'Calcicolae' group (comprising eight species), centred on the Agulhas Plain and containing three calcarenite-endemic species, *Aspalathus aciloba*, *Aspalathus calcarea* and *Aspalathus candidula*, the latter two species being sympatric and all three closely related. Phylogeographic analysis of the restioid genus *Thamnochortus* shows that a clade of mostly calcicolous species emerged on the Agulhas Plain; included from the coastal flora are the edaphically widespread *Thamnochortus erectus* as well as the calcicoles *Thamnochortus fraternus*, *Thamnochortus muirii*, *Thamnochortus paniculatus*, *Thamnochortus pluristachyus* and *Thamnochortus spicigerus* (*Linder & Mann, 1998*). Another example comes from *Metalasia*, in which all calcicoles—*Metalasia calcicola*, *Metalasia erectifolia*, *Metalasia luteola*, *Metalasia muricata*

and *Metalasia umbelliformis*—belong to the same clade that emerged during the Plio–Pleistocene (*Bengtson et al., 2014*; *Bengtson, Anderberg & Karis, 2014*). Other than the widespread, dune-endemic *M. muricata*, all other species are restricted to calcarenites on the Agulhas Plain. Thus, while limited, *in-situ* diversification has precipitated a few small clades in the coastal flora, and available information suggests that these are presently concentrated on the Agulhas Plain of the CFR.

*Otholobium* provides an interesting example of limited cladogenesis on calcareous substrata involving both ecological and geographic speciation. *Otholobium bracteolatum*, a dune-endemic species from the southern and southwestern CFR, is closely related to the calcarenite endemic *Otholobium sabulosum* from the Agulhas Plain, (*Stirton & Muasya, 2017*), suggesting that these two species have evolved due to differential selective pressures in the dune (more dynamic) and calcarenite (more stable) environment. *Otholobium* sp. nov. 'algoensis', a dune endemic from the southeastern CFR, is presumably sister to *Otholobium bracteolatum* (it was, until recently, included with this species) (*Stirton, 1986*), suggesting that these two species may have diverged and evolved along separate trajectories following the drowning of a previously contiguous distribution range along the Palaeo-Agulhas Plain. Other examples of such putative geographic speciation include: *Thamnochortus insignis* and the closely related, undescribed *Thamnochortus* sp. A (*Lubke & Bredenkamp, 2019*), known only from calcarenite outcrops at the mouth of the Sundays River (Algoa Bay) in the eastern CFR; *Pelargonium suburbanum* subsp. *bipinnatifidum*, which occurs on dunes west of Mossel Bay, and *P. s.* subsp. *suburbanum*, an endemic of the dunes between St Francis Bay and Algoa Bay.; and the previous example of *Maytenus procumbens* and *Maytenus lucida*, occurring in the eastern and western CFR, respectively.

The west–east breaks in distribution along the Cape south coast of the abovementioned species pairs mirrors the distribution of many calcicolous species with disjunct populations, including several members of the Aizoaceae (*Carpobrotus acinaciformis*, *Conicosia pugioniformis* subsp. *muirii*, *Mesembryanthemum vanrensburgii*), Fabaceae (*Indigofera tomentosa*, *Lotononis glabra*, *Psoralea repens*) and Restionaceae (*Elegia fenestrata*, *Elegia microcarpa*, *Elegia tectorum*, *Restio eliocharis*, *Restio leptoclados*). Some of these disjunctions are well known, for example that of *Acmadenia obtusata* and *Ficinia truncata*, species restricted to calcarenites on the Agulhas Plain in the west and inland of Algoa Bay in the east (*Taylor & Morris, 1981*); others we report on here for the first time, for example for *Cliffortia obcordata* and *Mesembryanthemum vanrensburgii*, both of which have disjunct populations between Still Bay in the west and the St Francis Bay–Algoa Bay area in the east (the eastern populations were recorded during our recent field surveys; see https://www.inaturalist.org/observations/27579755 and https://www.inaturalist.org/observations/20939032). While these south-coast disjunctions vary somewhat geographically, they are most consistent and pronounced along the Tsitsikamma coast (Fig. 1)—an area largely devoid of calcareous substrata. We would expect similar, though less marked, disjunctions along other steep, rocky coasts where coastal cliffs have hampered the accumulation of marine aeolianites, for example around the Cape Peninsula, Cape Hangklip and the Mossel Bay coast. Phylogenetic investigation of disjunct

coastal-endemic clades and species will shed further light on the timing of sea-level rise and consequent population fragmentation in the Cape coastal flora.

Unusual substrata provide a selective force for the evolution of neoendemic species, but alternatively could also provide a refuge from competition for palaeoendemics (*Bruchmann & Hobohm, 2014*). What is the case for the CFR coastal flora? The bulk of species are likely neoendemics, for example the calcicolous clade of *Metalasia* emerged during the Pleistocene (*Bengtson et al., 2014*; *Bengtson, Anderberg & Karis, 2014*), and *Gladiolus griseus*, a west-coast dune endemic, diverged from other species in the *Gladiolus carinatus* species complex during the Pleistocene, around 0.46 Ma (*Rymer et al., 2010*). One taxonomically isolated species, which is largely restricted to coastal dune forests of the southwestern CFR, is the small tree *Maurocenia frangula* of the Celastraceae. Its membership to a monospecific genus and its restricted range suggest that it is a relict of a warmer, wetter climate, and that it likely evolved long before the rest of the calcicole flora in the Cape. However, recent phylogenetic studies show that *M. frangula* is not part of a basally branching lineage and place it as sister to *Cassine peragua* (*Simmons et al., 2008*), suggesting that this species may be a more recent derivative within the genus *Cassine*.

## Evolutionary adaptations in the Cape coastal flora

What type of adaptations have evolved in coastal-flora endemics in the Cape? Reciprocal transplant experiments of calcicolous and calcifugous Proteaceae species from the Agulhas Plain showed higher mortality and reduced growth between substrata than on their native substrata, indicating a strong physiological adaptation to particular edaphic environments (*Newton, Cowling & Lewis, 1991*; *Mustart & Cowling, 1993*). These calcareous substrata, containing high levels of $CaCo_3$, typically demand from plants a tolerance to Fe and P deficiencies (*Lee, 1999*). Species endemic to calcareous substrata have thus overcome major physiological constraints, specifically related to nutrient acquisition from highly alkaline soils. In members of the Proteaceae from the Agulhas Plain, for example, root traits for phosphorous acquisition differ between calcicolous and calcifugous sister taxa (*Shane, Cramer & Lambers, 2008*).

Other adaptations relate to the typical coastal disturbances experienced by plants in the dune environment (*Hesp, 1991*), including: salt tolerance to endure salt spray and soil salinity (*e.g.*, members of the Amaranthaceae); increased root, shoot and rhizome development to endure sand burial (*e.g.*, the graminoids *Ehrharta villosa*, *Ficinia dunensis*, *Thinopyrum distichum*, and shrubs *Hebenstretia cordata*, *Morella cordifolia*, *Psoralea repens*); leaf roll (*e.g.*, *Moraea australis*), leaf indumentum (*e.g.*, *Arctotheca populifolia*, *Gazania rigens*), succulence and putative CAM photosynthesis (*e.g.*, members of the Aizoaceae) to endure aridity, high solar radiation and high temperatures; and reduced stature—pronounced in species that develop cushion growth forms (*e.g.*, the shrub *Achyranthemum sordescens*)—as an adaptation to strong and persistent winds.

Among woody species of tropical origin, there further appears to have been an evolution of shrubby descendants from arborescent ancestors (*Cowling, 1983*), evident in genera like *Cassine*, *Cussonia*, *Euclea*, *Diospyros*, *Olea*, *Maytenus*, *Rapanea*, *Robsonodendron* and *Searsia* (Fig. 11). In certain cases, these calcicolous shrubs can develop as geoxyles, forming

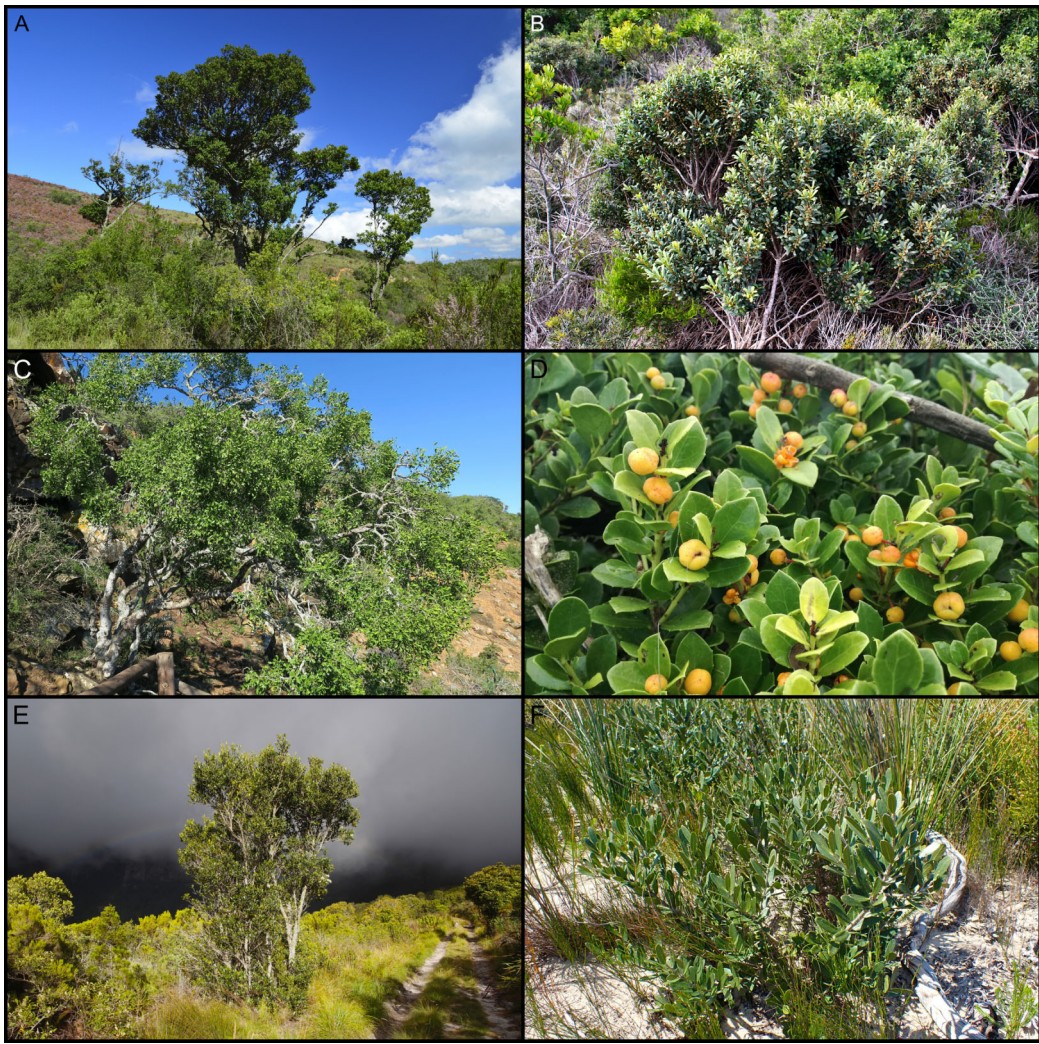

**Figure 11 Widespread trees of typically arborescent tropical genera and their calcicolous shrub descendants in the coastal flora of the Cape Floristic Region.** (A) *Rapanea melanophloes* is closely related to (B) the Cape dune endemic *Rapanea gilliana*. (C) *Maytenus undata* is the sister species of (D) the dune endemic *Maytenus procumbens*. (E) *Olea capensis*, which most frequently grows as a forest tree, is sister to (F) the Cape-endemic calcicole, *Olea exasperata*. Image credits: (A, B, D, F) B. Adriaan Grobler; (C) Graeme Pienaar, inaturalist.org 22021, licensed under CC BY NC; (E) Tony Rebelo, inaturalist.org 22021, licensed under CC BY 3.0 SA.

extensive networks of below-ground stems that produce aerial shoots along their length. Examples of such species include *Cassine peragua* subsp. *barbara*, *Diospyros pallens*, *Euclea racemosa*, *Olea exasperata*, *Rapanea gilliana* and *Searsia laevigata*. These geoxylic forms occur most frequently in dune landscapes hosting fire-prone fynbos vegetation (*Cowling, 1984*), and is likely an adaptation to recurrent fire (*Maurin et al., 2014*; *Lamont, He & Pausas, 2017*).

## The role of the Palaeo-Agulhas Plain

The CFR coastal flora is rich in species with high levels of geographic and edaphic endemism. Given the relative instability of coastal ecosystems, the strong edaphic selection

regime on calcareous substrata, as well as the small and fragmented areas covered by coastal dunes and calcarenites compared to other more widespread habitats in the CFR (*Grobler et al., 2020*), how did this remarkable diversity evolve? The answer likely lies in the glacial physiography of the Cape's south coast, where large tracts of calcareous substrata were exposed for long periods on the Palaeo-Agulhas Plain (PAP) (Fig. 1) during Pleistocene sea-level lowstands (*Cawthra et al., 2020*). At its maximum exposure (*e.g.*, during the Last Glacial Maximum—LGM), the PAP covered an area nearly equal to that of the contemporary CFR, although, edaphically, geologically and topographically, it presented a vastly different environment to the Cape of today (*Marean, Cowling & Franklin, 2020*). The subdued topography of this now-submerged plain provided ample space for the deposition of marine aeolianites: the area of coastal dunes on the PAP, for example, increased at least 38-fold compared to contemporary dune areas on the southern Cape coast, providing an additional 12–14,000 km$^2$ of habitat suitable for dune fynbos–thicket mosaics, while exposed Neogene calcarenites on the PAP, occupying ca. 19–20,000 km$^2$, increased the available habitat for limestone fynbos at least eight-fold (*Cowling et al., 2020*).

By invoking age-and-area theory, which explains much of the diversity patterns observed in the CFR (*Cowling et al., 2017*; *Forest, Colville & Cowling, 2018*; *Colville et al., 2020*), we argue that these expansive areas of calcareous habitat, exposed at length during Pleistocene glacials (*Jouzel et al., 2002*; *Waelbroeck et al., 2002*; *Fisher et al., 2010*), enabled the evolution of a rich coastal flora in the Cape (*Grobler et al., 2020*). Age-and-area theory posits that high levels of biodiversity amass in habitats characterized by sufficiently large areas to support viable biotic populations and by high environmental stability over evolutionary timescales, synergistically resulting in reduced extinction rates and increased speciation rates, and ultimately leading to the accumulation of species from both ancient lineages and more recent radiations (*Dynesius & Jansson, 2000*; *Jansson & Dynesius, 2002*; *Ricklefs, 2006*; *Fine, 2015*; *Schluter, 2016*). These same characteristics would lead to a high incidence of habitat specialists and range-restricted endemics (*Bruchmann & Hobohm, 2014*), as is the case in the CFR coastal flora.

Following the initiation of the Holocene, and during preceding late-Pleistocene interglacials, warming climates led to rapid increases in sea level (*Ramsay & Cooper, 2002*; *Murray-Wallace & Woodroffe, 2014*) and the subsequent drowning of large areas of coastal dunes along the Cape coast. This massive shrinkage and fragmentation of dunes relative to their historical (glacial) extents, which gave rise to their current configuration, would have had a major impact on the composition of the CFR dune flora and the population sizes of many of its component species. It is near certain that extinctions would have occurred, either through the gross inundation of range-restricted species populations, or because of severe genetic bottlenecks due to population fragmentation following sea-level transgressions. While calcarenite floras on the PAP would have experienced these same Pleistocene disruptions, the calcarenites exposed on the contemporary CFR coastal lowlands have not been subjected to sea-level transgression since the terminal Pliocene (*Partridge & Maud, 2000*; *Cowling, Procheş & Partridge, 2009*). This probably rendered these more inland calcareous substrata a refugium for the calcicole flora for the past

2.5 million years, especially along the Agulhas Plain in the western CFR, which was climatically buffered during Pleistocene glacials in comparison with calcarenite landscapes along the CFR's eastern margin (*Cowling et al., 1999*). The higher environmental stability of the western CFR, in concert with the relatively large and contiguous areas of calcarenite on the Agulhas Plain, likely contributed to the concentration of range-restricted calcicoles, especially calcarenite-endemic species, in this biogeographic centre.

## CONCLUSIONS

The Cape coastal flora is a distinctive, species-rich assemblage with high levels of edaphic and geographic endemism, comprising a significant proportion of all plant species in the CFR and with its endemic, calcicolous component representing 6% of the region's plant diversity—compared to other biodiversity hotspots, these are high tallies of unique plant biodiversity on calcareous substrata. The flora is a distinctly southern African formation, with most species belonging to Cape lineages (*sensu Linder, 2003*) and being endemic to the (G)CFR, although a considerable number of semi-desert and tropical lineages have speciated on calcareous substrata to produce species endemic to the region. Most of the endemic, calcicolous portion of the coastal flora emerged during the Plio-Pleistocene *via* ecological speciation upon colonization of novel calcareous substrata, with the ancient, calcifugous fynbos floras of montane habitats likely being the most significant source of lineages to the coastal flora. Of the two calcareous substrata occurring along the Cape coast, calcarenites appear to have been a more significant sink for Cape lineages than coastal dunes, suggesting that soil depth may be an important selective factor over and above the ecological filter presented by soil pH. Interestingly, the calcareous sands that host the coastal flora are home to few species that also grow in renosterveld, a vegetation type associated with relatively benign, neutral but heavier loamy soils, thus suggesting that soil texture may be an additional edaphic barrier to plant colonization in the Cape flora. These topics present a fertile subject for further research into the physiological and anatomical adaptations of calcicolous plant species in the Cape flora.

## ACKNOWLEDGEMENTS

We thank Charles H. Stirton, Ross C. Turner and Terry H. Trinder-Smith for sharing their taxonomic knowledge. Nick Helme and Douglas Euston-Brown are thanked for their comments on the edaphic distributions of certain species. Graeme Pienaar, Nick Helme, Ross C. Turner and Tony Rebelo are thanked for sharing their plant photographs. Our gratitude goes to Ruan van Mazijk and an anonymous reviewer for taking the time to critically read our manuscript; their comments and critique helped to improve the article.

### Funding

Funding for this research was provided by the National Research Foundation of South Africa (NRF) (Grant No. 110438). B. Adriaan Grobler was supported by an NRF

postdoctoral fellowship (Grant No. 116756). The funders had no role in study design, data collection and analysis, decision to publish, or preparation of the manuscript.

## Grant Disclosures

The following grant information was disclosed by the authors:
National Research Foundation of South Africa (NRF): 110438 and 116756.

## Competing Interests

Richard M. Cowling is an Academic Editor for PeerJ.

## Author Contributions

- B. Adriaan Grobler conceived and designed the experiments, performed the experiments, analyzed the data, prepared figures and/or tables, authored or reviewed drafts of the paper, and approved the final draft.
- Richard M. Cowling conceived and designed the experiments, performed the experiments, authored or reviewed drafts of the paper, and approved the final draft.

## Data Availability

The full list of plant species included in the coastal flora of the Cape Floristic Region, together with their traits, is available in the Supplementary File.

## Supplemental Information

Supplemental information for this article can be found online at http://dx.doi.org/10.7717/peerj.11916#supplemental-information.

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
