# Peer review of "The composition, geography, biology and assembly of the coastal flora of the Cape Floristic Region"

_PeerJ, doi:10.7717/peerj.11916_

## Round 0.1 · original submission · Minor Revisions

The reviewers suggested only few changes and I coincided with one of them in that if the checklist of the coastal flora of the Cape Floristic Region has not been published it will be worthy to include it in Supplementary material.

·

Basic reporting

The text is clearly and professionally written and meets all the major criteria of a good research paper, with clearly annotated and available raw-data. I've got no major comments, only some improvements to suggest below to polish off a very well presented study.

Some biogeographical terms being somewhat unclearly defined, especially from a non-specialist's perspective. I'd like to draw particular attention to "Afrotemperate" and "Afrotropical" (line 331, § Methods). These terms, while broadly intuitive, are potentially unclear as multiple geographical regions could be described as such. Does the Afrotemperate region, as far as this paper is concerned, refer to the somewhat more temperate southwestern corner of South Africa, or is it more synonymous with the pockets of Afromontane regions in the Cape, the Drakensberg and extending up towards eastern Africa? "Afrotropical" as used here could also be confusing, as this region is often considered to encompass all of Sub-Saharan Africa (when treated as one of the 8 or so global-scale biogeographic realms), and would then overlap entirely with any definition of the Afrotemperate/-montane region. Defining these terms more plainly, as has been done very helpfully for soil terminology (lines 370–373, § Methods), would make the text and the results themselves more apparent to non-specialist readers or readers from disciplines where these terms may have subtly different meanings.

The figures for this article are clear and insightful, but could be improved with slightly larger axis and label text, and some tweaks to the colour-schemes. In more detail, I have the following recommendations and comments (in descending order of importance):

1. Figure 6: I like this figure a lot, but it could be improved by making the range of circle-sizes greater (i.e., the biggest circle bigger, the smallest smaller), to make differences in the relative numbers of species more obvious. Ditto for the range of thicknesses in connecting lines. The raw species-counts for each circle could be annotated here for easy reference. Ditto for the no. shared species for the connecting lines. The colours of these circles don't add anything (though there's nothing with it, it is a bit distracting). Additionally, it would be useful to use the abbreviations for soil types used in this figure (CC, CS, SS, SH) in the main text as well, just to make the figure easier to use alongside the text.
2. Figures 4, 5: The data in these figures could be presented in a slightly more attention-grabbing way, perhaps. Maybe, seeing as it is mostly percentage/proportion data (4A–C, 5A,B), stacked-bars might simpler and make the data more self-evident? The colours in 4D and 5C are fine, but perhaps a different colour-scheme could be considered—maybe just grey and white? Also, throughout these figures in particular, the axis and labelling text could be bigger and thus more easy to read.
3. Figures 3, 7: Great figures—really creative and clear way to present these data! Although the colour helps to tell the squares apart, I did initially try to work-out what the colour represented for each family "as data". Perhaps, then, note in the caption that the colour is just for illustrative purposes in these figures, and does not represent any value or property of any of the families.
4. Figure 1: Elevation-colours could be more distinct from each other?

I feel these changes and improvements to the figures will help this paper really make its key findings and conclusions that much more apparent and stand-out.

Experimental design

The goal and knowledge was clearly defined. The authors set out to present a "sketch" of the history and features of this flora based on broad and comprehensive survey of its species, and they have done exactly that.

As mentioned above (see § Basic Reporting), the data are clearly annotated the collection process thereof is very well explained.

The data is notably supplemented by the authors' own observations, though the finer details of these observations are not presented. In principle this is not as transparent as it could be, but I doubt there are any problems with these observations, as this sort of data, along with expert-based data, is not unusual for this sort of study.

Validity of the findings

As mentioned above (see § Basic Reporting), the data are clearly annotated the collection process thereof is very well explained. Only fairly simple descriptive statistics were used in their analyses, and these are all perfectly sound.

The authors conclusions, following their insights and descriptions of the floristic data collated, are well substantiated and contextualised by previous literature and established ecological principles.

Additional comments

In its current manuscript-form, this article has a notably long discussion (22–23 pages) compared to the other sections (introduction: 4, study area and methods: 7, results: 7). While it is clear to me that the vast majority of the "conclusion-making" work and findings in this paper are made in the discussion, there are one or two parts of the discussions text that could go to the results section.

I have also made some smaller comments and suggestions on the text directly (see annotated review PDF).

To conclude, I think this article just needs a few adjustments for clarity's sake before being published, hence my recommendation for Minor Revisions. But, I must commend the authors nonetheless on a very impressive dataset and the conclusions they draw in this article.

Reviewer 2 ·

Basic reporting

This is a well-written paper, sufficiently referenced, and very comprehensive. The language and style are mostly OK (see minor comments for some word change suggestions). There are no loops left hanging or unanswered.

Experimental design

Overall, excellently done. The study focuses on existing data (in various forms) rather than (it seems) on novel fieldwork or sampling any poorly known parts of the study area. There is no problem with this, and the bulk of the methods described are literature-based. As a small addition, the authors might want to add a line that no structured fieldwork was undertaken as part of this study, and I would suggest that details relating to any specific fieldwork that the authors did do be included (perhaps including a table of dates, no. of specimens collected, where / how identified, and where deposited).

Validity of the findings

The findings are detailed in much detail and very comprehensively. It is a positive contribution to understanding the Cape Flora and its origins, with useful implications for conservation.

Unless I missed it, a full flora /checklist would be a useful addition to the paper as supplementary data.

Additional comments

Minor comments:
l.50 - replace "huge" with another word
l.51 - delete "only
ll.57-59 - Suggest deleting "Note that" and starting with "Most biologists"; insert "as per" after "limestone fynbos" and change citations to "Cowling & … (2001) and Rebelo et al. (2006)".
ll.135-140 - very long sentence, best break up.
l.286 - perhaps standardize sub-tropical / tropical use?
l.324-325 - would there have been challenges assigning as sp. to more than 1 biome?
ll.431-440 - rather replace "woodies" with something less colloquial.
ll725-726 - citation for this statement?
l.903 - spelling of "autocthonus" - should be "autochthonous"
ll.1022-1052 - similar situation in the Hantam-Roggeveld, with long-term climate stability (winter rainfall) allowing for edaphic speciation on dolerites, sandstones and shales.
l.1089 - "in-situ" is this correct for this journal? italics? (and also for "sensu" through the doc - italics?)
l.1223 - "led" should be "lead"

---

## Round 0.2 · accepted · Accept

Thank you for considering the suggestions and comments of the two reviewers. The figures and figure legends improved very much. The discussion is already lengthy, so I coincide with you in not extending this section. I think is a thorough analysis of the coastal flora of the CFR, congratulations again.